

# Warmer winter causes deepening and intensification of summer subsurface bloom in the Black Sea: the role of convection and self-shading mechanism

Elena A. Kubryakova, Arseny A. Kubryakov

Marine Hydrophysical Institute, Russian Academy of Sciences, Sevastopol, Russian Federation

*Correspondence to*: Elena A. Kubryakova (elena_kubryakova@mail.ru)

**Abstract.** Large differences in the vertical distribution of chlorophyll-a concentration (Chl) in a year with cold and warm winter are observed in the Black Sea on the base of Bio-Argo data. Stronger winter nutrient flux from deeper isopycnal layer in cold 2017 caused an increase of Chl in the upper 40-meter layer observed throughout the whole year – from February to October, with a

maximum exceeding 1.3 mg/m$^3$ in February-May of 2017. In warm 2016 with weaker winter convection maximum of Chl during winter-spring in this layer was only about 0.8-0.9 mg/m$^3$. However, the increase of Chl in 2017 led to strong light attenuation in the upper layer and a decrease of euphotic layer depth due to the "self-shading" mechanism. In 2016 with weaker bloom irradiance penetrated to a 40-70 m layer, below the maximum winter mixed layer depth (40-50 m) and reached the upper layer of nitroclyne, which was not affected by winter mixing. As a result, in warm 2016 the subsurface chlorophyll maximum deepens and Chl in

deeper layers was on 0.2-0.6 mg/m$^3$ higher than in 2017. The maximum difference (0.6 mg/m$^3$) was observed during a summer seasonal peak of irradiance due to the largest increase of light attenuation in 2017. As a result, the column-averaged yearly values of Chl in warm 2016 and cold 2017 were comparable. These results demonstrate that the effect of self-shading largely compensates the role of winter convective entrainment of nutrients and causes the deepening of Chl subsurface maximum in warmer years.

## 1 Introduction

Convective mixing is one of the most important mechanisms supplying nutrients in the euphotic layer in mid- and high latitudes (see e.g. Sorokin, 2002; Williams & Follows, 2003). Vertically entrained nutrients during winter are further consumed by phytoplankton and remineralized throughout the whole year. These nutrients, first, cause the early-spring bloom in the upper mixed layer (Sverdrup, 1953; Sorokin, 2002) and, further, can fuel the subsurface bloom in summer months (Williams & Follows, 2003; Kubryakova et al., 2018).

Particularly, in the Black Sea, the strongest winter-early spring bloom of diatoms (Mashtakova, 1985; Sorokin 2002; Mikayelyan et al., 2018) and early-summer bloom of coccolithophores (Mikaelyan et al., 2015; Silkin et al., 2014, 2019) are observed after severe winters. Several authors on the base of satellite data demonstrated that the variability of surface chlorophyll-a (Chl) on interannual time scales is correlated with winter sea surface temperature (Oguz et al., 2006; Finenko et al., 2014). Long-term analysis of in situ data (Mikaelyan et al., 2018) showed that winter severity significantly affects the taxonomic composition and

seasonal succession of phytoplankton in the Black Sea.

In a warm period of a year, Chl variability is characterized by so-called deep chlorophyll maximum (DCM) (Cullen, 2015), which position and strength have a strong variability in the Black Sea (Vedernikov, & Demidov, 1993; Finenko et al., 2002; Krivenko, 2010). According to the classical point of view phytoplankton occupies layer between the maximal amount of nutrients, situated in deeper layers, and light, increasing near the surface (Cullen, 2015). The nutrients entrainment in the productive layer at this time

is strongly related to the short-period mixing events (Williams & Follows, 2003) associated e.g. with storms (Iverson et al., 1974;



Zhang et al., 2014; Kubryakov, Zatsepin, et al., 2019), wind-driven or dynamic upwelling (McGillicuddy et al., 1998; Mikaelyan et al., 2020). Such physical processes provide vertical fluxes of nutrients in the lower part of the euphotic zone, which is one of the important reasons for phytoplankton grow in its subsurface maximum in summer (Jonhson et al., 2010; Cullen, 2015).

Another reason of the formation of DCM is the summer rise of solar radiation, which leads photoinhibition and decrease of Chl in the upper layer (Platt et al., 1982), widening of the euphotic layer, and deepening of the productive zone (Letelier et al., 2004; Mignot et al., 2014; Lavigne et al., 2015). Such displacement is partly associated with the increase of Chl in the cell caused by the photoadaptive behavior of phytoplankton (MacIntyre et al., 2002), documented for the Black Sea in (Finenko et al., 2002, 2005; Churilova et al., 2019).

Intense phytoplankton bloom causes strong light attenuation, as photosynthetic pigments strongly absorb the light (Morel, 1991). This effect known as self-shading can significantly impact on the phytoplankton growth in deeper layers, where the highest amount of nutrients is observed throughout the year (Shigesada & Okubo, 1981). The amount of Chl and related water clarity largely control the depth of the euphotic zone (Shigesada & Okubo, 1981; Morel, 1991) and, therefore, impact on the position of DCM (Letelier et al., 2004; Leach et al., 2018). The self-shading mechanism is one of the important factors driving the phytoplankton variability and taxonomic composition in the most productive waters, e.g. lakes, rivers, and coastal waters (Morel, 1991; Leach et

al., 2014; Churilova et al., 2020). On seasonal time scales, Letelier et al. (2004) have shown that this effect plays an important role in the variability of the euphotic layer, which additionally decreases in winter due to winter bloom of phytoplankton in the tropical Pacific Ocean. At the same time, the impact of self-shading on the interannual variability of the vertical distribution of Chl in the open ocean and, particularly, in the central Black Sea is mostly overlooked.

Recently-deployed Bio-Argo buoys give a possibility to obtain continuous measurements of Chl and light characteristics on the
interannual time scales with high vertical and time resolution (Claustre et al., 2010; Mignot et al., 2014). In our study, we use the measurements of these buoys in the Black Sea to investigate the effect of winter convection and irradiance on the vertical distribution of Chl in two years with contrasting winter conditions – warm 2016 and cold 2017.

## 2 Data and methods

The study is based on the bio-optical and hydrological measurements of two Bio-Argo buoys #6901866 and #7900591 located in
the deep part of the Black Sea in 2016 and 2017. The trajectories of the Bio-Argo buoys in the 2016-2017 period are shown in Fig. 1. The measurements of these buoys in both years were made in the central part and over the continental slope of the sea. We also note that the main results of this study do not change, if we analyze only data of buoy #7900591, which in the whole study period was situated in the east-central part of the sea.


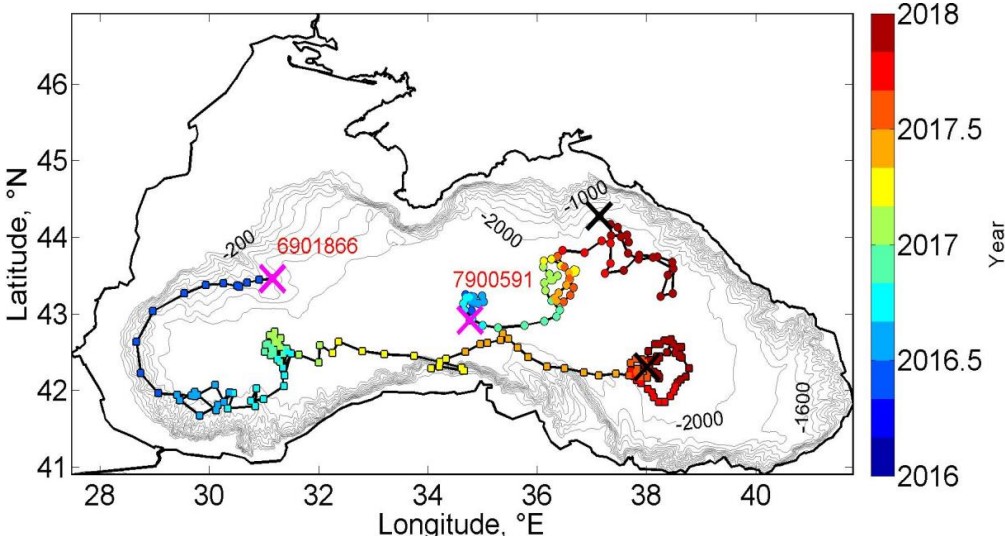

**Figure 1:** trajectories of the buoys #6901866 and #7900591 in 2016 and 2017. Colorbar shows the date of the measurements. Purple crosses – the start of the trajectories, black crosses –end of trajectories. Gray lines show the position of isobaths.

Data on the Chl and downwelling instantaneous irradiance ($E_d$, µmol photons m$^{-2}$ s$^{-1}$) integrated over 400–700 nm (PAR) were analyzed. Chl (mg m$^{-3}$) was retrieved from the chlorophyll fluorescence sensor of the Wetlabs ECO Triplet Puck using the retrieval equation described by Xing et al. (2011). Multispectral ocean color radiometer (OCR-504, SATLANTIC Inc.) was used to measure

the photosynthetic available radiation (PAR) in the water column. The precision of the device was 5 µmol photons m$^{-2}$ s$^{-1}$. Only $E_d$ measurements made near the local noontime (+/− 1 hour), which represent approximately the maximal daily downwelling irradiance, were analyzed.

Vertical gradients of $E_d$ were used to compute the diffuse attenuation coefficient K$_d(\lambda)$: $K_d(\lambda) = \ln\left(\frac{E_d(z+dz)}{E_d(z)}\right)/dz$, here z is depth, dz=1 m.

The Bio-Argo buoys provided data on salinity and temperature at the time of the bio-optical measurements. Mixed layer depth was estimated using a potential density threshold of 0.07 kg m$^{-3}$ (Kubryakov, Belokopytov et al., 2019). Bio-Argo buoy data have a high time resolution (1 and 5 days) and vertical resolution (1 m), is regular and is publicly available (Claustre et al., 2010).

### 3 Results

The Black Sea is characterized by several specific features, related mainly by the impact of large river inflow. Rivers cause its

strong haline stratification and bring a vast amount of organic matter (Vladimirov et al., 1997; Mankovsky et al., 2010), which reduces the transparency of its water. As a result, the diffuse attenuation coefficient in the sea is large (Organelli et al., 2017; Churilova et al., 2019) and the euphotic layer is relatively shallow, about 50 m (Vedernikov & Demidov, 1993). Due to the strong haline stratification the mixed layer depth also generally does not exceed 40-50 m in the central part of the sea and 50-70 m on its periphery and in anticyclonic eddies (Kubryakov, Belokopytov, et al., 2019). Due to these features, the upper border of the

nitroclyne is relatively shallow. It is situated approximately at 40-50 m depth and tightly coupled to isopycnals (isohalines) positons (Konovalov et al., 2005).

The results of this study are based on a comparative analysis of the bio-optical characteristics of the Black Sea in 2016 and 2017, which were characterized by different winter thermal conditions. In 2016, the winter was significantly warmer than in cold 2017





(Fig. 1a, b). The temperature in the upper 70-meter layer from January to March 2016 did not fall below 8.5°C (Fig. 2a), and its
column-averaged value varied from 8.5 to 9.0°C (Fig. 3, solid red line). Due to such warm conditions in winter of 2016 the cold
intermediate layer – a characteristic feature of the thermohaline structure of the Black Sea with a temperature of less than 8°C –
was absent this year (Fig. 2a). In 2017, the temperature in the upper 50-meter layer was on 1°C lower (Fig. 2b, Fig. 3, red dashed
line) as a result of severe winter cooling of the sea surface and convection. The minimum temperature in February reached 7°C on
the surface, and 7.5°C in the 0-70 m layer. As a result, the cold intermediate layer was observed in 2017 at 45-70 m in January -
September 2017.

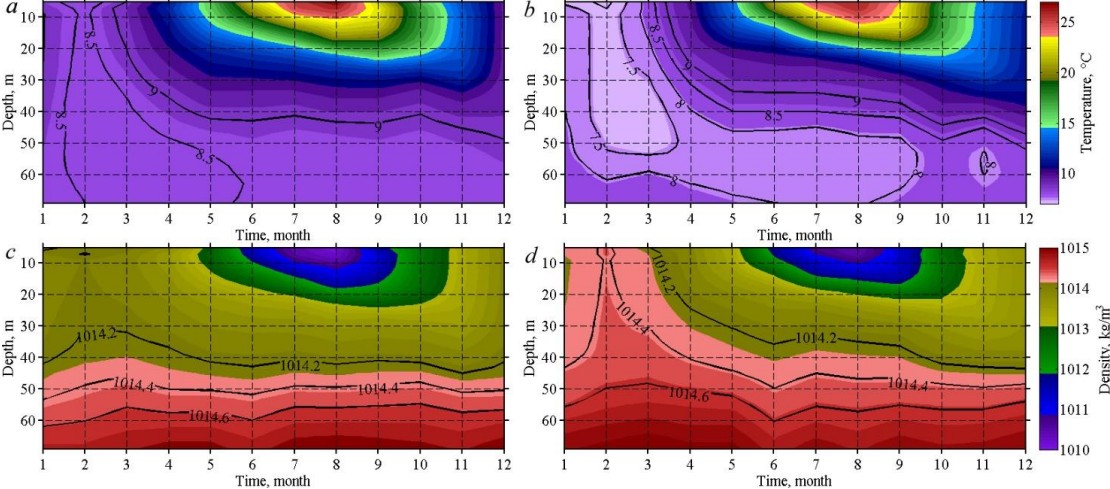

**Figure 2:** seasonal variability of temperature in 2016 **(a)** and 2017 **(b)**, density in 2016 **(c)** and 2017 **(d)**.

The decrease of temperature in 2017 led to an increase in the density of the surface layer and the entrainment of the underlying
isopycnal layer in the upper mixed layer (red line in Fig. 4a, b). Maximum entrainment was observed in February-March. In 2017,
intensive cooling caused an outcropping of isopycnal 1014.4 kg m$^{-3}$ on the surface. In 2016 the maximum density of the mixed
layer did not exceed 1013.8 kg m$^{-3}$ (Fig. 1c, d). Since the distribution of nutrients in the central part of the Black Sea is strongly
linked to the position of isopycnals, the density of the upper mixed layer is an indicator of the intensity of their entrainment
(Kubryakova et al., 2018). Thus, the cooling and entrainment of dense isopycnal layer (Fig. 1d) led to the fact that in 2017 a greater
number of nutrients entered the upper mixed layer as a result of winter mixing.

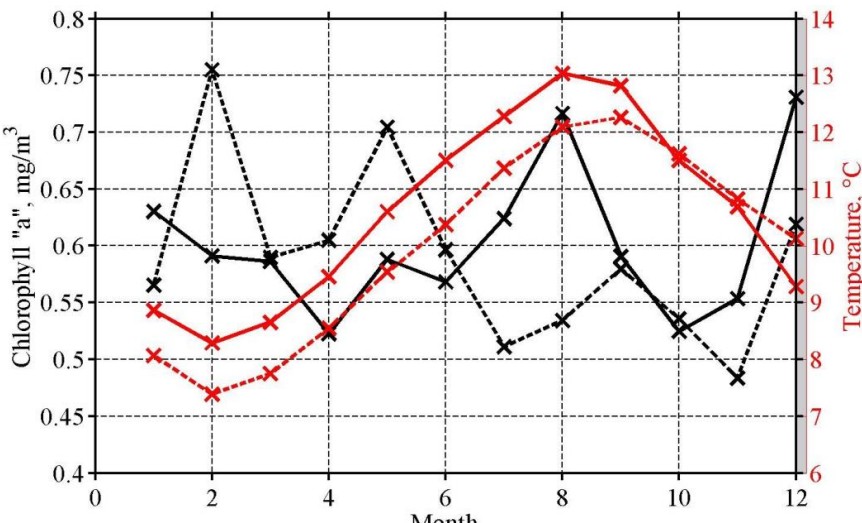

**Figure 3:** average seasonal variability of Chl and temperature in 2016 (solid line) and 2017 (dashed) in the 0-70 m layer.

The entrainment of nutrients in the upper layer caused the intense growth of phytoplankton, which is reflected in the increase in the Chl. In February 2017, the average Chl in 0-70 m layer was 0.75 mg m⁻³ (Fig. 3, black dashed line), and in the upper 30-meter

layer it exceeded 1.2 mg m⁻³ (Fig. 4b). In the same period of 2016, column-averaged Chl in the 0-70 m layer was on 0.15 mg m⁻³ lower (Fig. 3, black solid line) and was less than 0.9-1.0 mg m⁻³ in the 0-30 m layer (Fig. 4a).

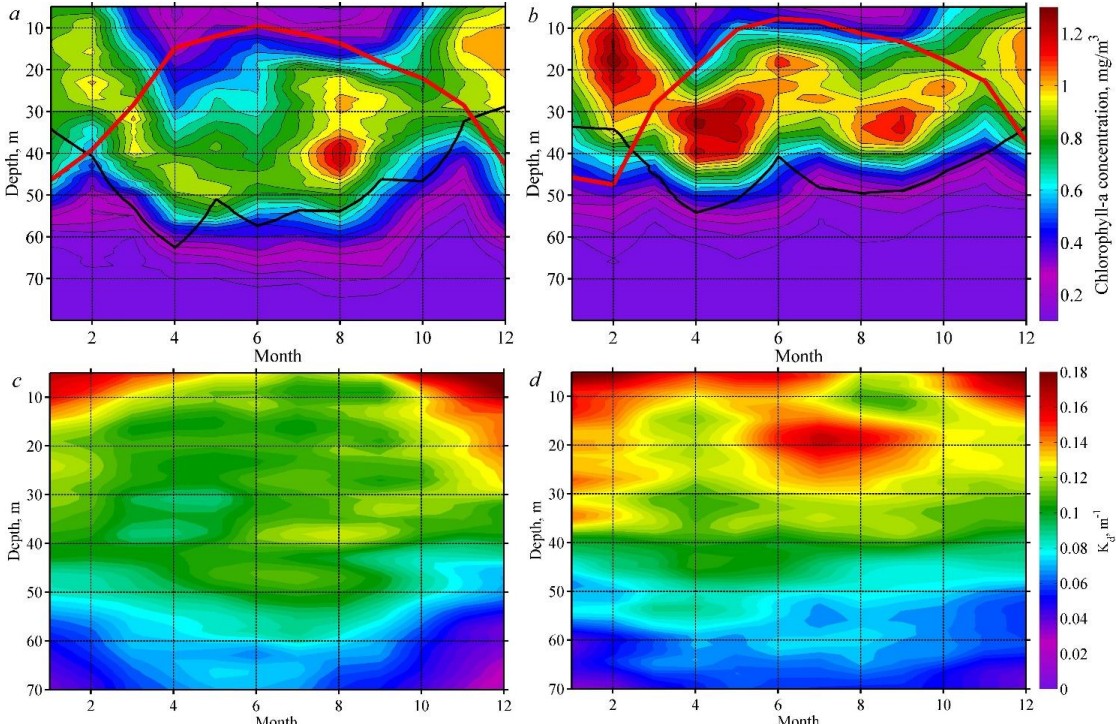

**Figure 4:** seasonal variability of Chl in 2016 **(a)** and 2017 **(b)**. The black line marks the border of the photic zone (maximum photosynthetic active radiation $E_d$=3 μmol photons m⁻² s⁻¹), the red line marks the mixed layer depth. Seasonal variability of diffuse attenuation coefficient $K_d$ (m⁻¹) in 2016 **(c)**, 2017 **(d)**.



Lower Chl values in 2016 in the 20-40 m layer were observed almost until the end of the year. From mid-March to July, the Chl in this layer in 2016 was about 0.8-0.9 mg m$^{-3}$, while in 2017 it was significantly higher – about 1.0-1.3 mg m$^{-3}$. Thus, intense entrainment of nutrients in the winter of 2017 led to an increase in biological productivity not only in winter but also in the

following months as a result of their remineralization. This effect, in particular, led to an unusually strong coccolithophore bloom in May-July 2017, which was observed from satellite measurements and Bio-Argo buoy data (Kubryakov, Mikaelyan, et al., 2019). At the same time, high concentrations of Chl (more than 0.5 mg m$^{-3}$) in 2017 were observed only in the 0-50 m layer, and in 2016 they were detected in a deeper layer, reaching 65 m. The changes in the vertical distribution of Chl are observed in the diagram of the Chl difference in 2017 and 2016 (Fig. 5a), which looked like "yin-yang sign". In 2017, from January to October, the upper 0-

40 m layer of Chl was larger than in 2016. The maximum difference was detected in March-June and reached 0.4-0.6 mg m$^{-3}$ in the 10-40 m layer. At the same time, Chl was significantly higher in the lower layers in 2016. Starting from March to November, the Chl values in the 40-60 m layer in warm 2016 were higher than in 2017 by the same value of 0.4-0.6 mg m$^{-3}$. The maximum excess of Chl in 2016 was observed in the summer period in the layer of 35-60 m, when its values were 0.5-0.6 mg m$^{-3}$ more than in 2017.


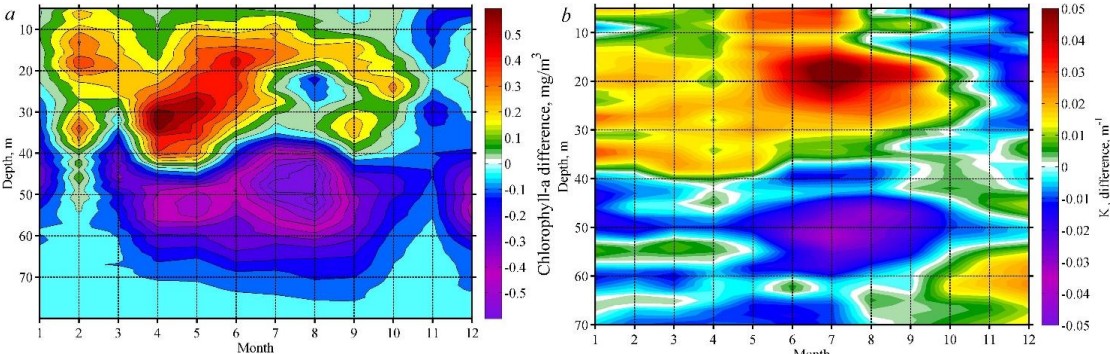

**Figure 5:** seasonal diagram of a difference of vertical distribution of Chl **(a)** and K$_d$ **(b)** between 2017 and 2016.

In Fig. 4a, b the black line marks the border of the photic zone is related to the isolume E$_d$=3 μmol of photons m$^{-2}$ s$^{-1}$. It can be seen that in 2017, this border was higher by 10 m in April, and by 20 m in June compared to 2016. This indicates that in 2017, the

growth of phytoplankton in the deep layer was limited by the low light conditions. The reason for this was an increase in the light attenuation coefficient K$_d$ in 2017 (Fig. 4c, d). The diffuse attenuation coefficient K$_d$ in the upper 30-meter layer from January to October 2017 exceeded 0.12 m$^{-1}$, reaching maximum values of 0.15 m$^{-1}$ in the summer period (Fig. 4d). At the same time, in 2016, the values of K$_d$ in February-October in the subsurface layer was about 0.1 m$^{-1}$ (Fig. 4c).

The diagram in Fig. 5b demonstrates the differences in K$_d$ distribution between 2017 and 2016. In 2017, the K$_d$ values in the 30-

meter layer were higher than in 2016 from January to October by 0.01-0.03 m$^{-1}$ (Fig. 5b). The largest differences (0.03-0.05 m$^{-1}$) were observed in June-September in the 15-30 m layer. At the same time, in the deep layer of 40-60 m, K$_d$ was lower in 2017 by 0.01-0.02 m$^{-1}$. The distribution of the K$_d$ difference diagram (Fig. 5b) is very similar to that for Chl (Fig. 5a), which indicates that the increase of phytoplankton and Chl in 2017 were one the main cause of the light attenuation due to self-shading mechanism.

Rise of K$_d$ led to a decrease in the PAR in the subsurface layers and shallowing of the euphotic zone. Diagrams in Fig. 6 demonstrate

PAR distribution in 2016, 2017, and their difference (2017-2016). It is well seen that in warm 2016 the decrease of production in the upper layers led to a larger amount of PAR in the subsurface layers of the sea (Fig. 6a). A significant increase of PAR by more than 1 μmol photons m$^{-2}$ s$^{-1}$ was observed in the entire layer of 0-60 m. Values of PAR were higher in February-March, June-August, and October-December 2016 than in 2017, when a much larger amount of light penetrated in the subsurface layer. In the





10-30 m layer, the values of PAR difference during the early spring bloom in February-April and autumn bloom in October-
November reached 20 µmol photons m$^{-2}$ s$^{-1}$ (Fig. 6c). In summer, these differences were greatest and exceeded 200 µmol photons
m$^{-2}$ s$^{-1}$ in the 0-25 m layer and were more than 10 µmol photons m$^{-2}$ s$^{-1}$ in the 40-meter layer. Taking the value of compensational
irradiance in the Black Sea as 3 µmol photons m$^{-2}$ s$^{-1}$ such PAR increase caused a significant widening of the euphotic zone.

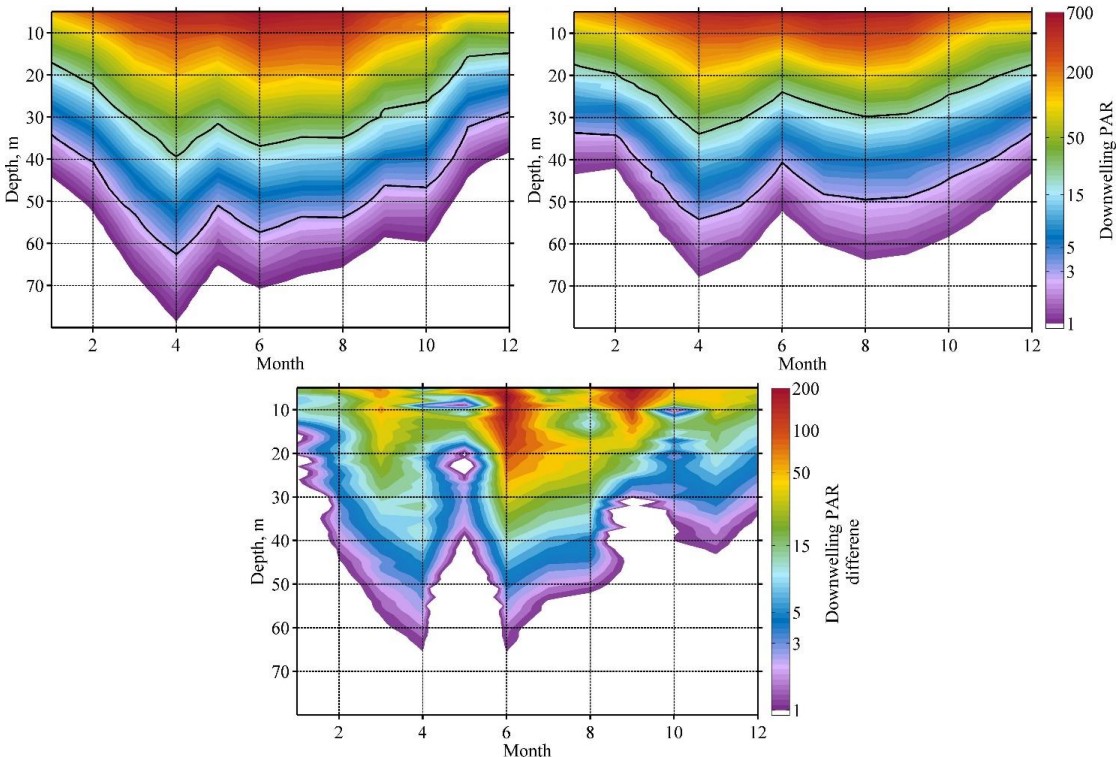

**Figure 6:** seasonal variability of PAR in 2016 **(a)**, 2017 **(b),** and PAR difference in 2017 and 2016 **(c)**. Black lines indicate the boundaries of the
photic zone (at the maximum photosynthetic active radiation $E_d$=3 µmol photons m$^{-2}$ s$^{-1}$ and $E_d$=25 µmol photons m$^{-2}$ s$^{-1}$).

As a result of light penetration, phytoplankton could develop in the deep layer (40-60 m). At the same time, the maximum winter
mixed layer depth (see Fig. 4a, b) did not exceed 45 m and winter convection did not reach these depths. Thus, the nutrients in the
45-60 m layer were not entrained in convective mixing and their concentration was undiluted by surface waters and stayed high.
The penetration of light into this layer in 2016 due to the decrease of self-shading caused the growth of phytoplankton directly in
the nitroclyne without mandatory mixing. A similar process is observed, in particular, in subtropical regions, where winter cooling
is almost absent. As a result, the largest rise of Chl in years with weak convection is observed in the deep layer and the subsurface
maximum of Chl deepens into layers unaffected by winter convection.

The increase of Chl in the deep layer in warm 2016 largely compensated its decrease in the upper layer caused by weak vertical
entrainment of nutrients (Fig. 4c). Despite the significantly larger Chl values in upper 40-meter layer in 2017, the difference in
column-averaged Chl over 0-70 m did not exceed 0.1 mg m$^{-3}$ in the first half of the year, except the period of intense convection
in February, when it reaches 0.15 mg m$^{-3}$ (Fig. 3). In summer, the sign of this difference changed to the opposite. Due to the rise
of light attenuation in the summer of 2017, the summer maximum of Chl was suppressed in this year. At the same time, in 2016,
light penetrated to larger depths, contributing to the development of a deep and wide subsurface maximum in August 2016 at 20-
60 m located close to the zone of high gradient of nutrients. In July-August, the column-averaged Chl in 2016 exceeded its values





in 2017 by 0.1-0.15 mg m$^{-3}$. Thus, the decrease in Chl in the first half of the year in the upper layer of 2016 due to the weak convective entrainment of nutrients in winter was compensated by its increase in the summer period due to the lack of self-shading effect and the development of a deep Chl maximum directly in the nitroclyne. Because of this, the yearly-averaged integral Chl in the euphotic layer in 2016 and 2017 became comparable (Fig. 3).

**4 Discussion**

The reasons for the variability of the characteristics of the deep chlorophyll maximum is one of the important and actively investigated oceanographic tasks (Cullen, 2015; Leach et al., 2018; Barbieux et al., 2019). Variability of its position and strength are related in different studies to the vertical distribution of nutrients (Hartman et al., 2014; Barbieux et al., 2019), optical characteristics of water and light availability (Morel, 1991; Mignot et al., 2014; Leach et al., 2018), density stratification (Navarro

& Ruiz, 2013). Our study approves that all these factors are important and provide a link between their impact on the vertical distribution of Chl on the example of the Black Sea based on continuous Bio-Argo measurements (see scheme in Fig. 7). Nitroclyne in the highly-stratified waters of the Black Sea is closely related to the isopycnals position (Konovalov et al., 2005). The increase of density of the upper mixed layer in winter leads to the convective mixing reaching the isopycnals with the same density. Deep isopycnal layer with a high amount of nutrients mixes with surface waters and defines the concentration of nutrients in the winter

mixed layer (0-40 m in the Black Sea) (Kubryakova et al., 2018; Mikaelyan et al., 2018; Silkin et al., 2019). Further thermal stratification stabilizes the water column but keeps the same concentration of nutrients. Rise of the irradiance causes the following spring growth of phytoplankton, which utilizes these nutrients. Part of them regenerates and another part sinks out to the nitroclyne. The regenerated nutrients further are used by the summer population of phytoplankton. For example, in the Black Sea, the strong coccolithophore bloom emerges after strong spring bloom of diatoms and their magnitude depends on the winter sea surface

temperature (Silkin et al., 2014; Mikaelyan et al., 2015). Thus, the phytoplankton concentration in the upper layer through the year will depend on the density of the preceding winter mixed layer.

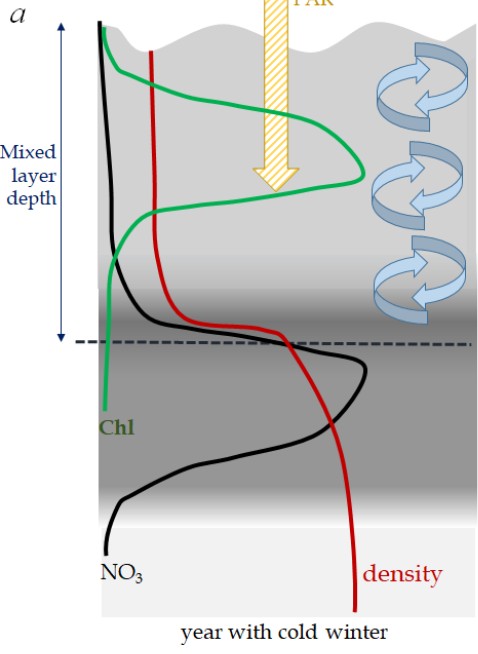

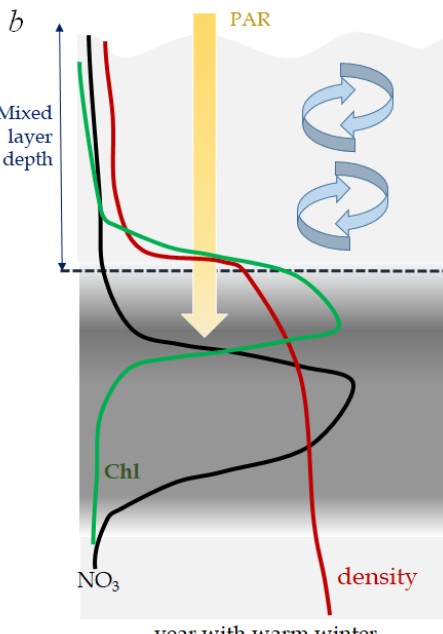





**Figure 7:** a scheme of the impact of convection and self-shading on the vertical distribution of chlorophyll-a. **(a)** In a year with cold winter, the larger amount of nutrients (grey color) is convectively entrained in the upper layer, which increases the growth of phytoplankton in the upper
layer and causes self-shading of deeper layer. Therefore, DCM moves to the upper layer. **(b)** In years with warm winter convective nutrient fluxes are low, the amount of phytoplankton and light attenuation decreases. In the summer period with the increase of PAR, light penetrates the upper layer of nitroclyne and causes intense and deep summer subsurface bloom. Therefore, the total amount of nutrients used by the phytoplankton in both years is comparable.

In the warm period of a year PAR strongly increases, euphotic layer deepens, and the surface layer becomes over-illuminated,
which is one of the possible reasons for the formation of deep chlorophyll maximum in summer (Platt et al., 1982; Mignot et al., 2014). The phytoplankton pigments, absorb light, and its variability largely define the optical characteristics of the water through the self-shading mechanism (Morel, 1991; Churilova et al., 2020). Thereby in the water with higher nutrient concentration (e.g. after cold winter), the light penetration to the deeper layer weakens. The water clarity decreases, which causes uplift of a productive layer or DCM to the surface (Mignot et al., 2014; Leach et al., 2018), as we observed in 2017 in the Black Sea (Fig. 3, Fig. 6a).
On the opposite, in the water with relatively low nutrient fluxes (as in 2016 in the Black Sea), the absence of self-shading promotes the light penetration to the deeper layer, which leads to the deepening and widening of the productive layer (Fig. 6b). Thereby DCM displacement is driven by the amount of nutrients and related self-shading by phytoplankton.

Light attenuation depends not only on the phytoplankton or Chl, but also on the attenuation by clear water and dissolved organic matter (DOM) (Morel, 1991). Independently of the phytoplankton concentration, the amount of light will additionally decrease in
the deeper layer. Particularly, in the Black Sea, the impact of DOM on light attenuation is very high due to the intense river discharge (Mankovsky et al., 2010; Organelli et al., 2017). The impact of this effect and lower nutrient concentration should lead to a decrease in the total amount of Chl after warmer years. However, in our study column-averaged Chl in years with high and low winter nutrient fluxes was almost equal, which should have some explanation.

Another source of light attenuation, which is nutrient-dependent, is autochthonous DOM formed due to the release of lipids during
lysis of phytoplankton cells. Particularly, in the Black Sea, the termination of the strong coccolithophores, probably as a result of viral lysis, provides a large amount of DOM in the upper layer (Kubryakov, Mikaeyan et al., 2019). This process was, particularly, important after very strong coccolithophore bloom in 2017 resulting in the observed maximum attenuation in July-August at 10-30 m depth (Fig. 5b). DOM release additionally increases the light attenuation, which plays a role in equalizing the total productivity in warm and cold years.

However, significantly more important is that in the warm year light penetrates at a deeper layer. In our case, it penetrates at 40-60 m in the Black Sea, where the upper border of nitrocline is located, below which concentration of nitrates sharply increases. In the Black Sea, the winter mixed layer is only about 40 m, and the underlying rich in the nutrients layer was unaffected by winter dilution. Therefore, due to the absence of self-shading in years with weak convection, the maximum of Chl is located close to the layer with a high concentration of nutrients, which will increase the biological productivity. This process will be most effective in
the summer because in this time of the year the phytoplankton is fueled mainly from below by the intense diapycnal mixing caused by storms (Iverson et al., 1974; Zhang et al., 2014; Chacko, 2017) or, e.g. eddy-driven upwelling (McGillicuddy et al., 1998; Mikaelyan et al., 2020). Particularly, in the Black Sea, several storms in August 2015 led to the subsurface maximum of Chl at the base of the euphotic layer reaching 7 mg m$^{-3}$ (Kubryakov, Zatsepin et al., 2019). Such nutrient fluxes depend on the gradient of nutrient concentration and will be larger in years with deeper bloom. Additionally, in the years with warm winter the cyclonic
circulation in the basin is usually weak, which intensifies the cross-shelf exchange in the Black Sea providing horizontal nutrient fluxes in the basin (Oguz et al., 2002; Kubryakov, Stanichny, et al., 2018). Thereby increase of nutrient fluxes in summer compensates their decrease in winter during years with weak winter convection and the differences in yearly averaged Chl in cold and warm periods are small.



On the other hand, the phytoplankton after cold and warm winter grows at different depths and, therefore, the environmental (light and nutrients) conditions change significantly. Such variability can be one of the reasons for the changes in seasonal succession of phytoplankton after cold and usual winters, studied in detail by Mikaelyan et al. (2018). In agreement with our analysis, authors show that after cold winter nutrients in upper layers significantly increase. This causes intense spring diatom bloom (Mashtakova, 1985; Sorokin, 2002; Silkin et al., 2014) which is followed by strong coccolithophore bloom (Mikaelyan et al., 2015). The lysis of coccolithophores causes the rise of DOM in the upper layer (Kubryakov, Mikaelyan, et al., 2019), which decreases the width thickness of the euphotic zone and the efficiency of the upward nutrient fluxes in it from the deeper layer. The release of DOM triggers the microbial loop, which transfers the trophic energy in smaller species. All these factors can explain the high biomass of diatoms in spring and low in summer after cold winters documented by Mikaelyan et al. (2018).

In contrast after warm winter phytoplankton in spring grows in low-nutrient conditions. Thereby in spring diatom biomass is lower and dinoflagellates biomass reaches maximum, according to (Mikelyan et al., 2018). However, in summer phytoplankton develops in the deeper layer with a higher amount of nutrients, which may explain the larger diatom biomass and diversity in warmer years documented by Mikelyan et al. (2018). Deepening of the euphotic layer may also promote the growth of species adapted to low light with low biomass and high Chl content in cells (Falkowski & La Roshe, 1991; MacIntyre et al., 2002; Latasa et al., 2017). Particularly, in the Black Sea, the lower border of the euphotic zone is characterized by the domination of small flagellates and unicellular cyanobacteria (Churilova et al., 2019; Mikaelyan et al., 2020), which may have an advantage in the years with warm winters.

## 5 Conclusions

A comparative analysis of the bio-optical characteristics of the Black Sea for 2016 and 2017 based on Bio-Argo buoys showed that the intensity of winter convection largely controls the bio-productivity and the position of the deep maximum Chl throughout the year:

- intensive water cooling in the cold winter of 2017 caused the involvement of deep waters in the upper layer of the sea, the growth of phytoplankton, and a significant increase in Chl in the upper layer. In the 0-40 m layer from March to October, the Chl values exceeded by 0.3-0.6 mg m$^{-3}$ the values in the same period of 2016. This increase in Chl led to an increase in the light attenuation coefficient and a decrease in the euphotic layer of the basin as a result of the self-shading effect. As a result, the Chl values in the 40-60 m layer in 2017 were significantly lower than in the warm 2016;

- in 2016, which was characterized by a warm winter and weak Chl involvement in the upper layer was much lower than in 2017. Due to the increased transparency of the water, a large number of PAR penetrated the nitrocline to a depth of 40-60 m, which was not affected by convective mixing. As a result, in the lower 30-60 m layer, Chl in 2016 exceeded its values in 2017 by 0.1-0.6 mg m$^{-3}$. Particularly significant difference was observed in the summer period, when the attenuation coefficient in the 10-30 m layer in 2016 was the lowest compared to 2017.

Thus, the decrease in Chl in 2016 in the spring in years with warm winters because of the weakening of convection involving nutrients compensated its growth in the deep layer, directly at the top of nitracline in the summer, due to the weakening of the effect of self-shadowing. As a result, the estimates of the integral layer of Chl were comparable in years with warm and cold winters. At the same time, the intensity of winter convection led to significant changes in the vertical distribution of Chl. In the year with warm winter, the subsurface maximum of Chl was shifted to the lower layer; in the year with cold ones, it was observed closer to the surface. Such light-driven inverse relation between winter and summer nutrient fluxes can be one of the important reasons for the observed interannual changes of seasonal succession in the ocean.



**Data availability**

Bio-Argo data were made freely available from IFREMER data archive [ftp://ftp.ifremer.fr/].

**Author contribution**

Elena Kubryakova and Arseny Kubryakov processed the data of Bio-Argo buoys, performed the analysis. Elena Kubryakova drafted the manuscript and designed the figures. Arseny Kubryakov was involved in planning and supervised the work.

**Competing interests**

The authors declare that they have no conflict of interest, no competing financial interests.

**Acknowledgments**

Analysis of the chlorophyll and light attenuation variability from Bio-Argo buoys is supported by the Russian Science Foundation (grant No. 19-77-00029). The study of the impact of PAR on the position of DCM is supported by the Russian Foundation for Basic Research (grant No. 20-05-00068). Data acquisition and processing were made with the support of Marine Hydrophysical Institute, Russian Academy of Sciences, under state assignment No. 0827-2019-0002.

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
