# Peer review of "Warmer winter causes deepening and intensification of summer subsurface bloom in the Black Sea: the role of convection and selfshading mechanism"

_Biogeosciences, 2020_

## Referee Comment (RC1) · Anonymous Referee #1 · 10 Jul 2020

Review on manuscript: BG-2020-210 entitled: "Warmer winter causes deepening and intensification of summer subsurface bloom in th Black Sea: the role of convection and self-shading mechanism"

Authors present 2-years time-series data from two Bio-Argo floats measuring temperature, salinity, Chl-a flurescence and irradiance in the Black Sea. They observed differences in deep chlorophyll maximum depth and intensity between summer 2016 and 2017. In 2016, DCM was deeper with lower maximum Chl-a concentration than in 2017. Authors explained these differences by previous winter conditions. Authors argue that if more nutrients are supplied in surface waters during winter, they can sustain during the whole summer period via remineralisation an higher phytoplankton biomass and a shallower DCM. This paper is interesting because it raises questions about which factors control DCM. As DCM results from an equilibrium between light (impacted by phytoplankton itself) and nutrients, determining which factor determines its position and intensity remain a challenging question. However, the authors presented a theory without giving the strong proofs and arguments. In fact although they claimed in the conclusion that they have showed that "the intensity of winter convection largely controls the bio-productivity and the position of the deep maximum Chl throughout the year", no strong differences is observed in Mixed Layer Depth (MLD) between 2016 (40m) and 2017 (45m), no data are provided about nutrient distribution. Finally, the impact of small scale structures such as fronts, eddies, etc. which are known to impact the nutrient vertical distribution and the DCM are ignored although the Argo floats trajectories indicates the presence of eddies or small gyres. Then, I recommend to reject this manuscript as it is.

However, given the value of the Bio-Argo dataset in this region and the interest of the scientific community about DCM, I would recommend to the authors to resubmit later their manuscript after major modifications and improvements. I advise to the author to add information about nutrients distribution in the Black Sea, to deeply reconsider the theory which are presented in this paper and to support any new theory with arguments and data. I also suggest to the authors to avoid the monthly averaging of the data. By this way small scale events can be considered. In addition, I would like authors investigate what happens in June 2017 when DCM unexpectedly uplifts.

Specific comments:

Introduction:

line 21-22, winter phytoplankton remineralisation needs reference. Same nutrients sustain spring bloom and summer DCM? You need to reference and argument.

line 25: Strong?

Line 29: "winter severity" is too general.

Line 32: DCM in Black Sea, please provide more details

Lines 39-43: photo-adaptation and photo-inhibition mechanisms need to be better described.

Line 56: Please replace the term "winter convection" by "winter mixing". Convection implies very deep mixed layers and specific mechanisms.

A presentation of the Back Sea with water mass presentation and circulation, nutrient and phytoplankton distribution is missing.

Data and Method:

Lines:62-63 please give details and show data. Regading Chl-a concentration data. How did you treat non photochemical quenching?

Line 74: Can you better justified your 0.07 kg/m3 criteria for MLD calculation. The criteria 0.03 or 0.05 kg/m3 are more commonly used. With a criteria of 0.07, MLD may be overestimated.

Results:

Lines 79-86: this paragraph should be in the introduction. Please provide concentration values for nitrate and phosphate in the Black Sea.

Line 93: "Convection" should be replaced by "mixing"

Line 94: what is the "cold intermediate layer"?

Figure 2: Please use a continuous color palette. These discontinuous colours can artificially emphasis differences in two situations which may be not so different (opposition between red and yellow colors).

Line 108: Without any data on nutrients concentration how can you argue that there is an entrainment of nutrients?

Figure 4: Can you explain the Chl-a increase in August 2016 and the DCM uplift in June 2017? Line 118-120: Please provide evidence to support this statement: "Thus, intense entrainmentof nutrients in the winter of 2017 led to an increase in biological productivity not only in winter but also in the following months as a result of their remineralization." Figure 5: Is this figure necessary?

Discussion

line 182: Indicate to which isopycnal nitracline is related and draw it on Figure 2.

line 185 : "Further thermal stratification stabilizes the water column but keeps the same concentration of nutrients", nutrients are generally rapidly consumed, data are needed to support this statement.

Line 188: Hypothesis on regeneration need to be support by strong data. In fact, regeneration generally happens in depth due to particles sedimentation.

Figure 7: This figure and the associated conclusions should be removed or at least deeply reviewed. Regarding the figure itself, it is very surprising to see a Chl-a DCM shape inside the mixed layer. In the mixed layer, one can expect homogeneous Chl-a profiles. Authors should have mentioned at least: "winter MLD" and "summer Chl-a vertical profile". Instead of the PAR arrows, it would be more accurate to indicate the position of isolumes as this information is available from Bio-Argo data. Regarding the theory explained in figure caption, I don't think the data presented in this paper allow to support it. Although authors didn't support with nutrient data the statement "In a cold winter, the larger amount of nutrients (grey color) is convectively entrained in the upper layer", my maine concern is for the following statement about case (b) : "In the summer period with increase of PAR, light penetrates the upper layer of nitroclyne and causes intense and deep summer subsurface bloom. Therefore, the total amount of nutrients

[Figure]

used by the phytoplankton in both year is comparable." In fact, summer light increase happens in both years and as soon as surface nutrients are consumed, DCM forms and deepens. It seems impossible that DCM production in oligotrophic conditions can compensate additional winter production permitted by extra nutrients inputs in surface waters. In addition, authors should remind that deep DCM have generally an higher Chl-a/biomass ratio than shallower DCM as Chl-a per cell increases to compensate the decreasing of light.

Line 200-207. Hypothesis on self-shading due to higher winter Chl-a concentration for explaining shallower DCM during the full summer season is doubtful. In fact, what which have been observed before is that as soon as bloom ends, DCM set up and deepens due to lower nutrient availability and higher light availability.

---

## Referee Comment (RC2) · Anonymous Referee #2 · 18 Jul 2020

This study investigates on the role of winter mixing and self-shading as drivers of the DCM position and magnitude in the Black Sea. Results are based on the analysis of a two-year time series of chlorophyll-a and photosynthetically available radiation (PAR) vertical profiles acquired by Biogeochemical-Argo floats. The analysis focuses on differences between a warm winter in 2016 and a cold winter in 2017. Self-shading by phytoplankton bloom has a main role in positioning the DCM. The paper cannot be published in the present form. Main criticisms are reported hereafter:

1) The role of nitrates is widely discussed though data have not been presented. The

role of NO3 is discussed using literature and MLD differences between 2016 and 2017. However, from Figure 4, MLDs look very similar. To support conclusions, NO3 profiles must be analysed. These profiles exist and have been acquired by the float 6901866. Other NO3 profiles have also been acquired in the same area by the float 6903240.

2) The difference between winters seems more related to mesoscale circulation than to a severe winter. In Figure 1, red squares clearly show that in winter 2017 the floats were entrapped within eddies, which would help explain also density features in Figure 2d. To support conclusion the impact of mesoscale should be addressed and, if any, excluded.

3) Self-shading is interpreted only as a change in chlorophyll concentration (derived from fluorescence measurements) which is used as a proxy of algal biomass or at least to indicate productive layers. Using only chlorophyll makes hard to establish if changes in the magnitude of DCM are due to actual changes in productive biomass because the same modifications could be due to photoacclimation. Optical measurements such as the optical backscattering coefficient could help decipher what changes in chlorophyll are due to actual modifications in algal biomass. Backscattering measurements are available for all the floats in the Black Sea.

4) Argo data analysis lacks of details, applied protocols and procedures and this makes difficult to evaluate its appropriateness. For example, for chlorophyll, it is unclear if NPQ at the surface and CDOM influence at depth have been corrected. How chl has been calibrated? Radiometry has been quality controlled before Kd computation?

Other comments below:

Line 15: 0.2-0.6 mg m3 is a large range of variability

Lines 17-18: to rephrase as it looks contracditory

Line 42: remove "in" before references

Line 59: why only these floats? More floats are available in the Black Sea

Line 60: the floats sampled a longer period. It would be interesting to see what's happened during winter in other years.

Line 69: which equation in Xing 2011 has been used? Have you calibrated Chl with Xing procedure? In this case, I suppose you have calibrated Chl using Ed(490). Ed(490) and derived Kd are quite correlated with PAR (see Morel et al., 2007 Remote Sensing of Environment) thus making analysis of the relationships between Chl and PAR as presented in this study not fully independent.

Line 69: how have you taken into account spikes and other source of errors in Chl profiles?

Line 70: please give the reference for 5 umol photons.

Line 73: have PAR profiles been quality controlled? Have you taken into account also clouds when computing Kd? Could it be possible to see Kd profiles? I imagine that they should be noisy as a 1 m window is very narrow.

Line 77: Claustre et al 2010 is not the right reference for the public database. Argo data are available at http://doi.org/10.17882/42182 or can be downloaded from the DAC (such as Coriolis).

Line 77: Temperature and salinity have been quality-controlled?

Figures 2, 4, 6: profiles have been monthly averaged? Details are missing to understand how single profiles have been managed before drawing the figures.

Line 81: replace "large" with "high"

Figures 2 and 3: add MLD

Line 105: to prove with data

Line 109: why average instead of integration over depth?

Figures 4c and d: add MLD and euphotic depth

Line 124: remove "yin yang sign"

Line 133: add reference for isolume 3 umol photons, why not using the 1% of irradiance just below the surface?

Line 134: shallower instead of higher?

Figures 5 and 6: I suggest to show PAR and Kd time series on the same figure

Caption figure 6: explain the choice of 25 umol photons

Figure 6: show MLD, DCM, euohotic depth

Figure 7 does not sum up your case study as in the figure MLDs are different. In this study the MLDs in winter look very similar.

Paragraph at Line 208: the influence of CDOM is less detectable from PAR as PAR integrates irradiance between 400 and 700 nm while the highest CDOM influence is in the UV range. As irradiance profiles at 380 nm are measured by BGC floats, Kd(380) should be computed to corroborate or not your statements.

Line 213: please give some explanations

Line 214: Are lipids colored?

Line 237: with no nutrient data this statement cannot be proved by this study

---

## Short Comment (SC1) · 26 Jul 2020

|                                                                                                                      | 1  | Seasonal stages of chlorophyll-a vertical distribution and its relation to the light conditions                                          |
|----------------------------------------------------------------------------------------------------------------------|----|------------------------------------------------------------------------------------------------------------------------------------------|
| 1                                                                                                                    | 2  | in the Black sea from Bio-Argo measurements                                                                                              |
| 2                                                                                                                    | 3  |                                                                                                                                          |
| 3
                                                                                                     | 4  | Kubryakov 1 A.A., Mikaelyan A.S. 2 , Stanichny S.V 1 , Kubryakova E.A. 1                     |
| 5
                                                                                                     | 5  | 1 Marine Hydrophysical Institute, Russian Academy of Sciences, 2, Kapitanskaya str.,                                          |
| 7
                                                                                                     | 6  | Sevastopol, 299011, Russia                                                                                                               |
| 9
                                                                                              | 7  | 2 Shirshov Institute of Oceanology, Russian Academy of Sciences, 36, Nakhimovski prosp.,                                      |
| 12                                                                                                                   | 8  | Moscow, 117997, Russia                                                                                                                   |
| 14                                                                                                                   | 9  | Abstract                                                                                                                                 |
| 15
                                                                                                   | 10 |                                                                                                                                          |
| 17
                                                                                                   | 11 | The year-to-year seasonal variability of the vertical distribution of chlorophyll-a (Chl), its                                           |
| 19
| 12 | relation with light conditions and mixing were studied in the Black Sea on a basis of 5-year (2014-                                      |
|                                                                                                                      | 13 | 2018) measurements of four Bio-Argo buoys. It was shown that the dependence of Chl on the                                                |
|                                                                                                                      | 14 | logarithm of the instantaneous Photosynthetically Available Radiation (PAR) at noon (Ed), has                                            |
|                                                                                                                      | 15 | quasigaussian shape. The majority of Chl values were in Ed interval from 3 to 330 $\mu$ mol photons                                      |
|                                                                                                                      | 16 | $m^{-2} s^{-1}$ with a maximum at 20 $\mu$ mol photons $m^{-2} s^{-1}$ . During all seasons the high values of Chl were                  |
|                                                                                                                      | 17 | located above the Ed of 3 $\mu$ mol photons m -2 s -1 or daily integrated PAR of 0.08 mole photons m -2 |
|                                                                                                                      | 18 | d -1 . This isolume can be regarded as a value close to compensational irradiance for the Black Sea                           |
|                                                                                                                      | 19 | phytoplankton restricting the productive layer. Low Chl observed at high Ed $> 330 \mu$ mol photons                                      |
|                                                                                                                      | 20 | m -2 s -1 evidences about important role of photoadaptaition and non-photochemical quenching in                    |
|                                                                                                                      | 21 | phytoplankton cells defining the amount of Chl in surface layer. In turn, Chl largely determines                                         |
|                                                                                                                      | 22 | light conditions in the deep layers by modulating attenuation coefficient of PAR (Kd) and the                                            |
| 39
                                                                                                   | 23 | dependence of Kd on Chl is well described by obtained in a study power function. In the study we                                         |
| 41
                                                                                                   | 24 | described the seasonal variability in vertical distribution of Ed and presented the monthly averaged                                     |
| 43                                                                                                                   | 25 | values of Kd in 5 m bins. Further, analysis of Bio-Argo data allows to distinguish 7 different stages                                    |
| 45                                                                                                                   | 26 | of Chl annual succession in the basin. They are: 1) winter minimum; 2) March early-spring peak;                                          |
| 46
                                                                                                   | 27 | 3) April-May depth-averaged minimum and the deepest Chl peak; 4) increased Chl in the                                                    |
| 48
                                                                                                   | 28 | thermocline in the second part of May and June; 5) large deep Chl peak in August; 6) early autumn                                        |
| 50
                                                                                                   | 29 | depth-averaged minimum in September-October; 7) late autumn-early winter bloom in November-                                              |
| 52
                                                                                                   | 30 | January. The roles of different abiotic and biotic factors (mixing, cross-shelf transport,                                               |
| 53
                                                                                                   | 31 | illumination, light attenuation, phytoplankton growth rate limitation, grazing) in the formation of                                      |
| 55
                                                                                                   | 32 | each of these stages are discussed.                                                                                                      |
| 57
                                                                                                   | 33 |                                                                                                                                          |
| 59
                                                                                                   | 34 | 1. Introduction                                                                                                                          |

One of the main characteristics of the marine ecosystem is the vertical distribution of chlorophyll-a concentration (*Chl*), which is a proxy of phytoplankton abundance and reflects the level of primary production in the basin (Demidov, 2009; Finenko et al., 2005, 2009; Yunev, 2011). Vertical variability of the phytoplankton characteristics in the Black Sea was analyzed basing on field data in a large amount of studies. Many of them were dedicated to the estimation of the season cycle of the phytoplankton biomass, *Chl* and primary production, regional features of these characteristics and their dependence on different physical factors (Sorokin, 1983; Zernova, Nezlin, 1983; Vedernikov, Demidov, 1993; Mikaelyan, 1997; Demidov, 1999, 2008; Berseneva et al., 2004; Finenko et al., 2005; Stelmah, 2006; Krivenko, 2010; Krivenko, Parhomenko, 2011; Mikaelyan et al., 2011, 2018; Silkin et al., 2018). It was shown that column-averaged Chl have generally three seasonal maximums (winter-early spring, summer, autumn) (Vedernikov, Demidov, 1993; Demidov, 1999; Finenko et al., 2005; Krivenko, 2010; Kubryakova et al., 2018), while surface *Chl* is characterized by two maxima in late autumn and early spring (Berseneva et al., 2004; Demidov, 2009; Krivenko, Parhomenko, 2010). General feature of Chl vertical distribution is deepening of peak during warm period of a year and a formation of a so-called deep chlorophyll maximum (Sorokin, 1983; Vedernikov, Demidov, 1993), similarly as in the other areas of the World Ocean at the same latitudes. The variability of the thickness, depth and shape of summer deep maximum in the Black Sea were investigated in details in (Finenko et al., 2005; Krivenko, 2010).

Stellite scanners provide a large amount of data about spatial-temporal variability of surface *Chl* (Kopelevich et al., 2002; Nezlin, 2008; Finenko et al., 2014; Kubryakov et al., 2016). Particularly, they give information about the importance of the impact of the mesoscale eddy dynamics on the cross-shelf fluxes of nutrients and *Chl* to the deep part of the basin (Oguz et al., 2002; Shapiro et al., 2009; Kubryakov et al., 2016). Satellite *Chl* data helped to reveal the key role of the temporal water stratification in the winter pulsing blooms of phytoplankton (Mikaelyan et al., 2017a). Modis satellite scanner measurements of *Chl* together with Bio-Argo data showed the strong impact of storms on the development of anomalous blooms of phytoplankton in the warm period of a year (Kubryakov et al., 2019a). Long-term analysis of satellite and in-situ data showed that seasonal surface peak of *Chl* is controlled by river discharge in the shelf areas and by start of winter convection in the deep areas of the basin (Krivenko, Parhomenko, 2010; Finenko et al., 2014; Kubryakova et al., 2018). Unfortunately, satellite data give information only about surface variability and they are unable to elucidate the important features of *Chl* seasonal dynamics, such as its subsurface maximum.

Although field measurements give results on the variability of the vertical distribution of *Chl*, usually, they have a rather rough resolution 5-15 meters. Having in mind a thin vertical structure of *Chl* in the ocean (Mankovsky et al., 2010; see review in Durham, Stoker, 2012), such low vertical resolution can lead to the smoothing of the main vertical features of phytoplankton distribution, especially in the zones of the sharp changes of *Chl* and physical characteristics, i.e. near thermocline or pycnocline.

Another important problem is a lack of simultaneous continuous measurements of light parameters (downwelling irradiance, diffuse attenuation) and phytoplankton characteristics. Although the data on the seasonal changes of the euphotic layer was obtained in several studies (Vedernikov, Demidov, 1993; Demidov, 1999; Finenko et al., 2002), the detailed joint analysis of vertical profiles of irradiance and *Chl* is almost absent in the Black Sea. Optical characteristics of the Black Sea waters were studied in several in-situ studies (Vladimirov et al., 1997; Churilova, 2004; 2008; 2009, 2017; Mankovsky et al., 2010). However, due to the lack of continuous measurements, especially in recent years, the data on seasonal variability in vertical distribution of light attenuation coefficients are very restricted.

Despite the relation of light, the mixed layer depth (MLD) and spring phytoplankton blooms was firstly described by (Gran and Braarud 1935; Sverdrup, 1953), until now there are large debates on some crucial questions. Particularly, recent findings reveal that the classical definition of the euphotic depth as 1% of surface Photosynthetically Available Irradiance (PAR) does not define the real thickness of the productive layer (Banse, 2004; Letelier et al., 2004). The primary production and high Chl can occur either higher or lower than this border (see e.g. Letelier et al., 2004; Mara et al., 2014). In the Black Sea, in several classical works the value of 0.1% of surface PAR instead of 1% was used for the definition of the photic zone (Vedernikov, Demidov, 1993; Demidov, 1999). Still, there are evidences that rather high amount of phytoplankton can be observed as just above, as below of this border (0.1% of surface PAR) (Demidov, 1999).

Another important question is the mechanism of the early spring phytoplankton bloom. Recently, studies have shown that the classical model relating the spring bloom to the mixed layer shallowing sometimes does not work, e.g. in the Northern Atlantic (Boss, Behrenfield, 2010; Ferrari et al., 2015). Instead, the authors showed that spring bloom starts when the wind turbulent mixing becomes negligible that stops mixing in the upper mixed layer, which thickness can be still large (Taylor, Ferrari, 2011; Ferrari et al., 2015). Further, in this article, we will show that in the Black Sea the different case is observed – spring bloom develops both inside and under the mixed layer, and its vertical expansion is controlled, first of all, by an increase in downwelling irradiance.

Another issue, which was only briefly mentioned in the Black Sea (Sorokin, 1980; 103 Vedernikov, Demidov, 1993) is the impact of the high insolation on the *Chl* seasonal dynamics in 104 the upper layers, through photoinhibition mechanisms. Photoadaptive behavior of phytoplankton 105 plays the important role in its seasonal variability, especially in summer months (Finenko et al., 2002, 2005; Churilova et al., 2009, 2017). This photoinhibition/phodoadaptaion can be crucial for
the *Chl* dynamics in the upper layers, as it was demonstrated recently for the South Ocean
(Alderkamp et al., 2010, 2011; Xing et al., 2019).

Therefore, despite large previous efforts there are still significant gaps in the current understanding of the variability of *Chl* and its relation to the physical factors, especially irradiance on seasonal time scales. One of the important reasons of these gaps is the absence of continuous measurements of the biooptical properties. The recent deployment of the Bio-Argo floats gives a possibility to overcome this problem (Claustre et al., 2010). Bio-Argo buoys today gives simultaneous information about several key biooptical properties of the ocean, including downwelling irradiance, Chl, light backscaterring and others on regular basis and with exceptionally high vertical resolution. These data allowed to make several important insights on the phytoplankton dynamics in different areas of the World Ocean (see reference list in (http://biogeochemical-argo.org/).

In 2014-2015 several Bio-Argo floats were released in the Black Sea. The measurements of these Bio-Argo recently were used to study processes on the oxic-anoxic interface, oxygen variability, particle distribution and diapycnal mixing in the Black Sea (Stanev et al., 2013, 2017, 2018; Capet et al., 2016). In (Kubryakova et al., 2018) seasonal dynamics of depth-averaged Chl were estimated on the base of Bio-Argo data. In (Kubryakov et al., 2019a) the response of the vertical distribution of Chl on several storms was described. Patterns of the seasonal and interannual variability of vertical distribution of *Chl* were briefly discussed in (Kubryakov, Stanichny, 2016; Stanev et al. 2017) and the relation of *Chl* and light attenuation at 490 nm was obtained in Organelli et al., 2017. (Callieri et al., 2019) use this data to investigate the phenomena of "deep red fluorescence" in the basin. In (Kubryakov et al., 2019a) these measurements were used for a detailed study of the vertical evolution of summer and winter coccolithophore blooms in the Black Sea, their relationship with Chl variability and the effect on the formation of dissolved organic matter. However, to date, detailed studies of Chl variability related to changes in hydrophysical and optical characteristics conducted on a basis of Bio-Argo data were absent.

In this study we use the 5-year (2014-2018) high-resolution Bio-Argo measurements of *Chl* and PAR to investigate the seasonal variability of *Chl* in different years, its relation with light characteristics and other physical factors. In the first part of the paper we consider the seasonal variability of vertical distribution of PAR, its impact on the *Chl*, and the relation between Chl and PAR attenuation. Further, the seasonal succession of *Chl* in the Black Sea is divided on seven different stages: winter minimum, early spring maximum, April-May minimum, two summer subsurface maximums, early autumn decrease of *Chl* and late autumn *Chl* peak. The roles of different abiotic and biotic factors (mixing, cross-shelf transport, grazing) on the development ofeach of these stages are discussed.

**2. Data and methods**

The study is based on data of 4 Bio-Argo floats measuring continuously the biooptical characteristics in the Black Sea during 2014-2018 period. Data were downloaded from IFREMER data archive (ftp://ftp.ifremer.fr/). Most of the time the buoys were located over the continental slope or in the deep basin (depth > 1000 m) (fig.1). The vertical resolution of Bio-Argo data was 1 m, the temporal resolution varied from 1 to 5 days.

---

## Author Comment (AC1) · 26 Jul 2020

We would like to thank the reviewer for comments and for valuable and constructive suggestions for improving the paper.

Comment #1:

"Authors present 2-years time-series data from two Bio-Argo floats measuring temperature, salinity, Chl-a flurescence and irradiance in the Black Sea. They observed differences in deep chlorophyll maximum depth and intensity between summer 2016

and 2017. In 2016, DCM was deeper with lower maximum Chl-a concentration than in 2017. Authors explained these differences by previous winter conditions. Authors argue that if more nutrients are supplied in surface waters during winter, they can sustain during the whole summer period via remineralisation an higher phytoplankton biomass and a shallower DCM. This paper is interesting because it raises questions about which factors control DCM. As DCM results from an equilibrium between light (impacted by phytoplankton itself) and nutrients, determining which factor determines its position and intensity remain a challenging question. However, the authors presented a theory without giving the strong proofs and arguments. In fact although they claimed in the conclusion that they have showed that "the intensity of winter convection largely controls the bio-productivity and the position of the deep maximum Chl throughout the year", no strong differences is observed in Mixed Layer Depth (MLD) between 2016 (40m) and 2017 (45m), no data are provided about nutrient distribution".

Answer #1. Unfortunately, there is no direct information about newly entrained nitrates in the upper layers in the winter season.

There are some important reasons for it. The entrained nutrients are usually rapidly consumed and then are transformed into organic form – i.e. phytoplankton, zooplankton, dissolved organic matter, etc. To account for these entrained nutrients we need to know all the compounds where e.g. nitrogen is situated, which is almost impossible nowadays. Particularly, in our institute, we made several surveys with nitrates measurements included in 2016 and 2017 in the summer and autumn periods. However, this is certainly not enough to estimate nitrates coming in the euphotic layer continuously in short-period events of winter mixing throughout all autumn-winter season.

The regular optical-based nitrates measurements of Bio-Argo buoys could be a good alternative for this task. Unfortunately, in the Black Sea data of Bio-Argo buoys is poorly consistent with information of nitrates distribution known from numerous in-situ studies. In particular, Bio-Argo buoys show the persistent presence of more than 3 $\mu$M nitrates in the upper layer of the Black Sea throughout the year (see diagram in Fig. R1-left

in attached file), which is not consistent with 0.5 $\mu$M documented in many previous studies (Konovalov, Murray, 2001; Turgul et al., 2015). A possible reason for this is the complex optical characteristics of the Black Sea with a lot of dissolved organic matter, etc (see e.g. Organelle et al., 2017).

Therefore, we use indirect estimates of newly entrained nitrates.

- First, the winter of 2017 was one of the most severe in the Black Sea and this fact was already documented in several recent studies (Stanev et al., 2019; Capet et al., 2020). It was significantly colder than in warm 2016 and cause significantly stronger vertical mixing than in 2016 (Stanev et al., 2019; Capet et al., 2020). It is worth noting, that in colder winters, convection will be stronger and more nutrients will be entrained in the upper layer, than in warm winter. Please see the review in (Williams & Follows, 2003). For the Black Sea this is proven by the strong relationship between winter temperature and interannual variability of winter-early spring bloom of diatoms (Mashtakova, 1985; Sorokin 2002; Mikayelyan et al., 2018) and following the early-summer blooms of coccolithophore (Mikaelyan et al., 2015; Silkin et al., 2014, 2019), the variability of surface chlorophyll-Ðř (Chl-a) (Oguz et al., 2006; Finenko et al., 2014).

In the strongly stratified Black Sea, the depth location of nutricline is tightly coupled to certain isopycnals as it is shown in many chemical studies (Tugrul et al., 1992; Konovalov et al., 2005). That is why nutricline variations in $\sigma$-coordinates are significantly less than in-depth coordinates (Tugrul et al., 1992; Konovalov et al., 2005). The multi-annual vertical profiles of nitrate (NO3) and phosphate (PO4) presented in $\sigma$-coordinates for October, the month preceding the onset of intense winter convection, are shown in Fig. R1-right. For example, the concentration of nitrates begins to gradually increase below the isopycnal of 1014 kg/m3, and increases more sharply below the isopycnal of 1014.4 kg/m3 where the upper part of nutricline is located (Konovalov, Murray, 2001). The deeper isopycnals the winter convection reaches, the more new nutrients will be entrained into the euphotic layer. The tight relation between density and the position of chemical elements (see Konovalov et al., 2005) suggests that the

density of the upper mixed layer in winter can be used as a proxy, showing from which layers nutrients were entrained to the surface layer (Kubryakova et al., 2018).

At the same time, the mixed layer depth in the cold period of a year may vary significantly due to the dynamical forcing, such as eddies, large-scale circulation, etc (see in detail (Kubrykov et al., 2019)). This is related to the deepening of the density barrier – the main halocline. For example, in anticyclones, it can reach 100 m. However, if the density of the mixed layer remains low, then no new nitrates will be entrained from deep isopycnals layers.

The density of the mixed layer depends partly on the vertical uplift of isopycnals during the intensification of cyclonic circulation. The rise of cyclonic circulation on the opposite decreases mixed layer depth. Therefore, in the Black Sea the MLD is not correlated with sea surface temperature (Titov, 2004), but strongly depends on dynamic forcing (Kubryakov, Belokopytov, et al., 2019).

That is why the density rather than the depth of the mixed layer is a more robust indicator of the vertical entrainment of nutrients in winter. We use this indicator to show that in cold 2017 more nutrients are entrained in the euphotic layer than in warm 2016.

We extended the explanation in the revised version of the manuscript.

- Second, Chl-a is one of the widely-used indicators of the phytoplankton biomass, which directly depends on nutrient concentration. In 2017 Chl-a in winter and spring was higher than in 2016, which is consistent with the fact, that the winter convection and related vertical entrainment of nutrients was more intense in 2017.

We also want to underline that we are not basing on the quantitative values of nitrates, but use the above indicators to argue that in the cold winter of 2017 the vertical entrainment of nitrates was higher than in the warm winter of 2016. The increase of nutrient concentration in the Black Sea in the cold years was documented in the chemical study of (Tugrul et al., 2015).

Comment #2

"Finally, the impact of small scale structures such as fronts, eddies, etc. which are known to impact the nutrient vertical distribution and the DCM are ignored although the Argo floats trajectories indicates the presence of eddies or small gyres. Then, I recommend to reject this manuscript as it is. However, given the value of the Bio-Argo dataset in this region and the interest of the scientific community about DCM, I would recommend to the authors to resubmit later their manuscript after major modifications and improvements. I advise to the author to add information about nutrients distribution in the Black Sea, to deeply reconsider the theory which are presented in this paper and to support any new theory with arguments and data. I also suggest to the authors to avoid the monthly averaging of the data. By this way small scale events can be considered. In addition, I would like authors investigate what happens in June 2017 when DCM unexpectedly uplifts".

Answer #2. The investigation of the impact of small scale structures such as fronts, eddies on Chl-a is a very interesting and important problem. However, in this study, we are focused on the annual time scales. Particularly, Fig. 5 shows that in cold 2017 Chl-a in upper layers was higher in all seasons, while in warm 2016 it was higher in deeper layers in all seasons. This fact was observed during all investigated periods of both buoys measurements. Please see Fig. 5a, which is the main figure for this manuscript. That is, yearly average profiles of Chl-a are of main interest and they depend on the intensity of winter convection (and see Fig. R2 and R3 in the attached file).

The short-period variability of the Chl-a is out of the scope of this paper, but we briefly discuss it in the discussion part. Short period variability of the Chl-a in the summer period is related to the occasional entrainment of nutrients from nitrocline in the euphotic zone caused by storms or dynamical forcing (studied in the Black Sea by Kubryakov, Zatsepin et al. (2019)), such as eddies horizontal and vertical advection (see e.g. Oguz et al., 2002; Shapiro et al., 2010; Kubryakov et al., 2016). After warm winters with higher water transparency, the euphotic layer is deeper and closer to the nitroclyne. Therefore, we might expect that the impact of dynamics features in summer will be more effective in years with weak winter convection.

In Fig. R2 (in the attached file) we show the diagram of 5-days averaged profiles of Chl-a for both buoys in 2016 and 2017 to demonstrate that in both years the short-period variability takes place. It is also well-seen that these two buoys were situated in different dynamic features and the Chl-a variability differs among the buoys in both years. At the same time, it is visually seen that both buoys show that in warm 2016 Chl-a subsurface maximum was deeper than in cold 2017, which is the main conclusion of the study. It is also well seen that Chl-a was higher in 2017 in the winter-spring period in upper layers and higher in summer of 2016 in deep layers (35-55 m depth) (see Fig. R2, bottom). We will add this information in the revised version of the manuscript.

Specific comments (SC)

SC1: "line 21-22, winter phytoplankton remineralisation needs reference. Same nutrients sustain spring bloom and summer DCM? You need to reference and argument".

Answer SC1. We wrote this phrase more accurately:

"With the rise of stratification and irradiance vertically entrained nutrients during winter are further consumed by phytoplankton, which causes the early-spring bloom in the upper layers (Sverdrup, 1953; Sorokin, 2002). After the bloom, part of nutrients in organic form sinks out to the nitroclyne and another part regenerates, which can fuel the phytoplankton bloom in the warm period of the year (Williams & Follows, 2003). In the Black Sea according to (Lebedeva and Vostokov, 1984; Karl and Knauer, 1991) only a small fraction (~10%) of particulate flux is exported to deeper anoxic part of the sea. The most intense winter-early spring bloom of diatoms (Mashtakova, 1985; Sorokin 2002; Mikayelyan et al., 2018) and following the early-summer bloom of coccolithophores (Mikaelyan et al., 2015; Silkin et al., 2014, 2019) in the Black Sea are observed after severe winters, both of which are related to the entrained in winter nutrients. Long-term analysis of in-situ data (Mikaelyan et al., 2018) showed that winter temperature significantly affects the taxonomic composition and seasonal succession of phytoplankton in the Black Sea throughout the whole warm period of the year. Several authors on the base of satellite data demonstrated that the variability of surface chlorophyll-Đř (Chl) on interannual time scales is correlated with winter sea surface temperature (Oguz et al., 2006; Finenko et al., 2014). The biomodelling study of (Kubryakova et al., 2018) also shows that the intensity of the summer deep phytoplankton maximum also depends on the winter convection".

SC2: "line 25: Strong?"

Answer SC2. Changed to most intense. There are at least two blooms of diatoms in the Black Sea – in spring and autumn.

SC3: "Line 29: "winter severity" is too general".

Answer SC3. Changed to "winter temperature".

SC4: "Line 32: DCM in the Black Sea, please provide more details".

Answer SC4. We added a short description of DCM in the Black Sea. General feature of Chl-a vertical distribution is deepening of its peak during the warm period of a year and a formation of a so-called deep chlorophyll maximum at 15-50 m depth (Sorokin, 1983; Vedernikov, Demidov, 1993), similarly as in the other areas of the World Ocean at the same latitudes. The variability of the thickness, depth, and shape of summer DCM in the Black Sea was investigated in detail by Finenko et al. (2005), Krivenko (2010).

SC5: "Lines 39-43: photo-adaptation and photo-inhibition mechanisms need to be better described".

Answer SC5. We slightly extended the description in this part of the introduction. Also, we note that the investigation of these important processes generally are not the goal of this study. The possible impact of the photoadaptation on the ratio of Chl-a and biomass is shortly addressed in the Discussion: "Deepening of the euphotic layer may

also promote the growth of species adapted to low light with low biomass and high Chl content in cells (Falkowski & La Roshe, 1991; MacIntyre et al., 2002; Latasa et al., 2017). Particularly, in the Black Sea, the lower border of the euphotic zone is characterized by the domination of small flagellates and unicellular cyanobacteria (Churilova et al., 2019; Mikaelyan et al., 2020), which may have an advantage in the years with warm winters".

SC6: "Line 56: Please replace the term "winter convection" by "winter mixing". Convection implies very deep mixed layers and specific mechanisms".

Answer SC6. We partly agree with this comment and change the phrase to "winter mixing", because both wind, dynamic and cooling impact on the mixing. However, we must note that among these three factors, winter convection plays a most important role, and there are a lot of studies dedicated to the investigation of the winter convection, which is very important in the Black Sea:

Ivanov, L. I., Backhaus, J. O., Özsoy, E., & Wehde, H. (2001). Convection in the Black Sea during cold winters. Journal of marine systems, 31(1-3), 65-76.

Stanev, E. V., Roussenov, V. M., Rachev, N. H., & Staneva, J. V. (1995). Sea response to atmospheric variability. Model study for the Black Sea. Journal of Marine Systems, 6(3), 241-267.

Staneva, J. V., & Stanev, E. V. (1997). Cold intermediate water formation in the Black Sea. Analysis on numerical model simulations. In Sensitivity to Change: Black Sea, Baltic Sea and North Sea (pp. 375-393). Springer, Dordrecht.

and others.

Convection is driven by density differences in the fluid, e.g. the sinking of cold, dense water formed in winter. It can be deep or shallow, depending on stratification, which is very strong in the Black Sea.

SC7: "A presentation of the Back Sea with water mass presentation and circulation,

nutrient and phytoplankton distribution is missing".

Answer SC7. We added a short description of these features of the Black Sea in the text.

SC8: "Lines: 62-63 please give details and show data".

Answer SC8. We added the figures with the variability of both buoys in Fig. R2 (in the attached file).

SC9: "Regading Chl-a concentration data. How did you treat non photochemical quenching?"

Answer SC9. We use the standard product downloaded from ftp://ftp.ifremer.fr/ifremer/argo. The non-photochemical quenching was not corrected. We can expect that this effect will not alter the obtained result, as we focus on the differences of Chl between two years. NPQ is important in the most upper layers (0-15 m), while the differences in this study were observed in all 10-70 m layer (see Fig. 5). Also, NPQ depends primarily on the irradiance conditions on the surface, in which seasonal variability (change from summer to winter) is more or less uniform in both years. Therefore, we believe the correction of NPQ will not generally change the main results of the paper.

SC10: "Line 74: Can you better justified your 0.07 kg/m3 criteria for MLD calculation. The criteria 0.03 or 0.05 kg/m3 are more commonly used. With a criteria of 0.07, MLD may be overestimated".

Answer SC10. The criteria 0.03 is usually used globally (e.g. de Boyer Montégut et al., 2004). The criterion 0.07 used in this paper is regional and it was chosen exactly for the Black Sea. It was justified in our previous paper (Kubryakov, Belokopytov, et al., 2019), where we used composite analysis (see Fig. 1 (Kubryakov, Belokopytov, et al., 2019)) to show that this criterion is reliable for the Black Sea.

SC11: "Lines 79-86: this paragraph should be in the introduction. Please provide

concentration values for nitrate and phosphate in the Black Sea".

Answer SC11 We moved this paragraph to the introduction according to Your advice.

SC12: "Line 93: "Convection" should be replaced by "mixing".

Answer SC12. Replaced.

SC13: "Line 94: what is the "cold intermediate layer"?"

Answer SC13. We added a short description of the Cold Intermediate Layer (CIL) in the Introduction. The Cold intermediate layer is the layer of minimal temperature (T<8°C) situated at 50-150 m depth. During winter convection cold waters do not penetrate through the halo-pycnocline and form the CIL with a high amount of oxygen, which is further observed during the whole year. See also

Staneva, J. V., & Stanev, E. V. (1997). Cold intermediate water formation in the Black Sea. Analysis on numerical model simulations. In Sensitivity to Change: Black Sea, Baltic Sea and North Sea (pp. 375-393). Springer, Dordrecht.

Korotaev, G. K., Knysh, V. V., & Kubryakov, A. I. (2014). Study of formation process of cold intermediate layer based on reanalysis of Black Sea hydrophysical fields for 1971–1993. Izvestiya, Atmospheric and Oceanic Physics, 50(1), 35-48. And others.

SC14: "Figure 2: Please use a continuous color palette. These discontinuous colours can artificially emphasis differences in two situations which may be not so different (opposition between red and yellow colors)".

Answer SC14: We corrected the figure (see Fig. R4 in the attached file).

SC15: "Line 108: Without any data on nutrients concentration how can you argue that there is an entrainment of nutrients?"

Answer SC15. Please, see the answer to comment #1 above.

SC16: "Figure 4: Can you explain the Chl-a increase in August 2016 and the DCM

uplift in June 2017?"

Answer SC16: As it is stated in the answer on Comment #2 this paper is focused on the annual time scales and shows generally that in years with weak winter convection (mixing) the DCM is situated deeper all over the year (see Fig. 5), which is related to the effect of self-shading. In Fig. R2 (in the attached file) we show a diagram of "raw" profiles Chl-a variability for both buoys in 2016 and 2017 to demonstrate that in both years the short-period variability mainly controls the Chl-a dynamics. The observed short-period oscillations of Chl-a can be caused by numerous reasons. One of them is the storm-driven mixing in the warm period of the year (see Kubryakov et al., 2019), which provides the nutrient fluxes to the deep layer in the euphotic zone. As it is discussed in the paper this process will be more effective if the euphotic zone is situated deeper, as in 2016 (due to the absence of self-shading). Another important reason for chlorophyll variability in summer is the impact of early-summer coccolithophore blooms, which are very strong in the Black Sea (Mikaelyan et al., 2015). These blooms also depend on winter convection. They, particularly, cause an intense release of DOM during their termination (see Kubryakov, Zatsepin et al., 2019), which cause significant light attenuation and shallowing of the euphotic zone. Both these effects are described in the discussion. Mesoscale or large-scale circulation also can impact on the vertical displacement of the DCM. We added this comment to the study.

SC17: "Line 118-120: Please provide evidence to support this statement: "Thus, intense entrainment of nutrients in the winter of 2017 led to an increase in biological productivity not only in winter but also in the following months as a result of their remineralization.""

Answer SC17. The evidence is that the Chl-a (which is the indicator of biological productivity) was higher throughout the year in the upper layers of 2017. Additional evidence is the very intense coccolithophore blooms observed in May-July 2017 (Kubryakov, Mikaelyan, et al., 2019), which intensity also depends on winter convection, as it is shown in (Burenkov et al., 2011; Mikaelyan et al., 2011, 2015). We added

the latter comment to the study.

SC18: "Figure 5: Is this figure necessary?"

Answer SC18. This is the most important figure. Please, see the answer to comment #2 above.

SC19: "line 182: Indicate to which isopycnal nitracline is related and draw it on Figure 2".

Answer SC19. We added the graph of the nutrient distribution in the Black Sea to the paper (see Fig. R1 in the attached file). As it is seen the concentration of nitrates begins gradually increase below the isopycnal of 1014.0 kg/m3, and increases more sharply below the isopycnal of 1014.4 kg/m3 where the upper part of nutricline is located (Konovalov, Murray, 2001).

SC20: "line 185: "Further thermal stratification stabilizes the water column but keeps the same concentration of nutrients", nutrients are generally rapidly consumed, data are needed to support this statement".

Answer SC20. We corrected this phrase. Further thermal stratification stabilizes the water column. Entrained in winter period nutrients and the rise of the irradiance causes the following spring growth of phytoplankton.

SC21: "Line 188: Hypothesis on regeneration need to be support by strong data. In fact, regeneration generally happens in depth due to particles sedimentation".

Answer SC21. As it is stated in the introduction the impact of the winter entrainment of nutrients on the Black Sea phytoplankton was shown in many previous studies. "After the bloom, part of nutrients in organic form sinks out to the nitroclyne and another part regenerates, which can fuel the phytoplankton bloom in the warm period of the year (Williams & Follows, 2003). In the Black Sea according to (Lebedeva and Vostokov, 1984; Karl and Knauer, 1991) only a small fraction (∼10%) of particulate flux is exported to deeper anoxic part of the sea. The most intense winter-early spring bloom of

diatoms (Mashtakova, 1985; Sorokin 2002; Mikayelyan et al., 2018) and following the early-summer bloom of coccolithophores (Mikaelyan et al., 2015; Silkin et al., 2014, 2019) in the Black Sea are observed after severe winters, both of which are related to the entrained in winter nutrients. Long-term analysis of in-situ data (Mikaelyan et al., 2018) showed that winter temperature significantly affects the taxonomic composition and seasonal succession of phytoplankton in the Black Sea throughout the whole warm period of the year. Several authors on the base of satellite data demonstrated that the variability of surface chlorophyll-Đř (Chl) on interannual time scales is correlated with winter sea surface temperature (Oguz et al., 2006; Finenko et al., 2014). The biomodelling study of (Kubryakova et al., 2018) also shows that the intensity of the summer deep phytoplankton maximum also depends on the winter convection.

SC22: "Figure 7: This figure and the associated conclusions should be removed or at least deeply reviewed. Regarding the figure itself, it is very surprising to see a Chl-a DCM shape inside the mixed layer. In the mixed layer, one can expect homogeneous Chl-a profiles. Authors should have mentioned at least: "winter MLD" and "summer Chl-a vertical profile". Instead of the PAR arrows, it would be more accurate to indicate the position of isolumes as this information is available from Bio-Argo data".

Answer SC22: We agree and changed the Fig. 7: we added the position of euphotic zone and changed the captions on winter mixing and summer Chl-a (see Fig. R5 in the attached file).

SC23: "Regarding the theory explained in figure caption, I don't think the data presented in this paper allow to support it. Although authors didn't support with nutrient data the statement "In a cold winter, the larger amount of nutrients (grey color) is convectively entrained in the upper layer".

Answer SC23. Please see the answer to comment #1. Shortly, in cold winter the entrainment should be stronger. Chl-a is an indirect indicator of the amount of entrained nutrients.

SC24: "... my maine concern is for the following statement about case (b) : "In the summer period with increase of PAR, light penetrates the upper layer of nitroclyne and causes intense and deep summer subsurface bloom. Therefore, the total amount of nutrients used by the phytoplankton in both year is comparable." In fact, summer light increase happens in both years and as soon as surface nutrients are consumed, DCM forms and deepens".

Answer SC24. Yes, in summer light increase and DCM deepens. Important is to answer, why in summer of warm years light penetrates deeper than in summer of cold year? In more transparent waters (modulated by the low amount of nutrients), light reaches larger depths and DCM deepens stronger.

SC25: "It seems impossible that DCM production in oligotrophic conditions can compensate additional winter production permitted by extra nutrients inputs in surface waters".

Answer SC25. The Black Sea is mesotrophic, not oligotrophic. It has several specific features – a shallow mixed layer and nitrocline, etc. That is why the discussed effects can be more prominent in the Black Sea. However, they should work in any other regions of the World Ocean, where convection reaches nitrocline. The logic is simple:

– cold winter -> more nutrients are entrained in winter -> more phytoplankton (and DOM) -> less transparent waters -> lesser penetration of light -> shallower DCM;

–warm winter ->less nutrients are entrained in winter -> less phytoplankton (and DOM) -> more transparent waters -> deeper penetration of light -> light reaches deeper layer, where nutrient concentration is higher -> deeper DCM -> DCM is closer to nitroclyne.

In this study, this compensation is confirmed by the same value of column-averaged Chl-a in the warm and cold year (Fig. R3). We also added to the manuscript figure of the average annual profile of Chl-a in 2016 and 2017 (see Fig. R3).

SC26: "In addition, authors should remind that deep DCM have generally an higher

Chl-a/biomass ratio than shallower DCM as Chl-a per cell increases to compensate the decreasing of light".

Answer SC26. It is briefly written in the discussion at lines 247-250. Deepening of the euphotic layer may also promote the growth of species adapted to low light with low biomass and high Chl-a content in cells (Falkowski & La Roshe, 1991; MacIntyre et al., 2002; Latasa et al., 2017). Particularly, in the Black Sea, the lower border of the euphotic zone is characterized by the domination of small flagellates and unicellular cyanobacteria (Churilova et al., 2019; Mikaelyan et al., 2020), which may have an advantage in the years with warm winters.

SC27: "Line 200-207. Hypothesis on self-shading due to higher winter Chl-a concentration for explaining shallower DCM during the full summer season is doubtful. In fact, what which have been observed before is that as soon as bloom ends, DCM set up and deepens due to lower nutrient availability and higher light availability".

Answer SC27. Both of these sentences are true. But we need to explain higher light availability in a warm year. It is caused by the absence of the phytoplankton in the upper layers (no self-shading), which increases the transparency of the waters. This is directly shown in Fig. 5 and Fig. 6.

Please also note the supplement to this comment: https://www.biogeosciences-discuss.net/bg-2020-210/bg-2020-210-AC1-supplement.pdf
* * *
[Figure]

[Figure]

**Fig. 1.** Fig. R1: left– interannual diagram of NO3 ($\mu$M) measured by Bio-Argo buoy; right– multiannual averaged vertical profiles of NO3 ($\mu$M, black line) and PO4 ($\mu$M, red line) for October shown in $\sigma$-coordinate

[Figure]

**Fig. 2.** R2.Time variability of Chl by Bio-Argo buoy measurements #7900591(top) and buoy #6901866(central) in 2016(left) and 2017(right).Bottom–differences between 2017,2016 by buoy #6901866(left), #7900591(ri

[Figure]

**Fig. 3.** Fig. R3. Average profile of Chl-a in 2016 and 2017 by the measurements of the buoy #6901866 (left) and buoy #7900591 (center) and both buoy measurements (right).

[Figure]

[Figure]

**Fig. 4.** Fig. R4. Seasonal variability of temperature in 2016 (a) and 2017 (b), density in 2016 (c) and 2017 (d).

**a**

PAR

Winter
mixing

Winter
mixing

Euphotic
layer

summer Chl

NO$_3$

year with cold winter

**b**

PAR

Winter
mixing

Euphotic
layer

summer Chl

NO$_3$

year with warm winter

**Fig. 5.** R5.

---

## Author Comment (AC2) · 26 Jul 2020

We would like to thank the reviewer for comments and for valuable and constructive suggestions for improving the paper.

Comment #1

"The role of nitrates is widely discussed though data have not been presented. The role of NO3 is discussed using literature and MLD differences between 2016 and 2017. However, from Figure 4, MLDs look very similar. To support conclusions, NO3 profiles

must be analysed. These profiles exist and have been acquired by the float 6901866. Other NO3 profiles have also been acquired in the same area by the float 6903240".

Answer #1. Unfortunately, there is no direct information about newly entrained nitrates in the upper layers in the winter season. There are some important reasons for it.

The entrained nutrients are usually rapidly consumed and then are transformed into organic form – i.e. phytoplankton, zooplankton, dissolved organic matter, etc. To account for these entrained nutrients we need to know all the compounds where e.g. nitrogen is situated, which is almost impossible nowadays. Particularly, in our institute, we made several surveys with nitrates measurements included in 2016 and 2017 in the summer and autumn periods. However, this is certainly not enough to estimate nitrates coming in the euphotic layer continuously in short-period events of winter mixing throughout all autumn-winter season.

The regular optical-based nitrates measurements of Bio-Argo buoys could be a good alternative for this task. Unfortunately, in the Black Sea data of Bio-Argo buoys is poorly consistent with information of nitrates distribution known from numerous in-situ studies. In particular, Bio-Argo buoys show the persistent presence of more than 3 $\mu$M nitrates in the upper layer of the Black Sea throughout the year (see diagram in Fig. R1-left in the attached file), which is not consistent with 0.5 $\mu$M documented in many previous studies (Konovalov, Murray, 2001; Turgul et al., 2015). A possible reason for this is the complex optical characteristics of the Black Sea with a lot of dissolved organic matter, etc. (see e.g. Organelle et al., 2017).

Therefore, we use indirect estimates of newly entrained nitrates.

- In colder winters, convection will be stronger and more nutrients will be entrained in the upper layer, than in warm winter. Please see the review by Williams & Follows (2003). For the Black Sea this is proven by the strong relationship between winter temperature and interannual variability of winter-early spring bloom of diatoms (Mashtakova, 1985; Sorokin 2002; Mikayelyan et al., 2018) and following the early-summer

bloom of coccolithophores (Mikaelyan et al., 2015; Silkin et al., 2014, 2019), the variability of surface chlorophyll-Đř (Chl-a) (Oguz et al., 2006; Finenko et al., 2014). In the strongly stratified Black Sea, the depth location of nutricline is tightly related to certain isopycnals as it is shown in many chemical studies by Tugrul et al. (1992), Konovalov et al. (2005). That is why nutricline variations in $\sigma$-coordinates are significantly less than in-depth coordinates (Tugrul et al., 1992; Konovalov et al., 2005). The multi-annual vertical profiles of nitrate (NO3) and phosphate (PO4) presented in $\sigma$-coordinates for October, the month preceding the onset of intense winter convection, are shown in Fig. R1 (right) (in the attached file). For example, the concentration of nitrates begins to gradually increase below the isopycnal of 1014.0 kg/m3, and increases more sharply below the isopycnal of 1014.4 kg/m3 where the upper part of nutricline is located (Konovalov, Murray, 2001). The deeper isopycnals the winter convection reaches, the more new nutrients will be entrained into the euphotic layer. The tight relation between density and the position of chemical elements (see Konovalov et al., 2005) suggests that the density of the upper mixed layer in winter can be used as a proxy, showing from which layers nutrients were entrained to the surface layer (Kubryakova et al., 2018).

At the same time, the mixed layer depth in the cold period of a year may vary significantly due to the dynamical forcing, such as eddies, large-scale circulation, etc (see in detail (Kubryakov et al., 2019)). This is related to the deepening of the density barrier – the main halocline. For example, in anticyclones, it can reach 100 m. However, if the density of the mixed layer remains low, then no new nitrates will entrain from deep isopycnals layers.

The density of the mixed layer depends partly on the vertical uplift of isopycnals during the intensification of cyclonic circulation. The rise of cyclonic circulation on the opposite decreases mixed layer depth. Therefore, in the Black Sea the MLD is not correlated with sea surface temperature (Titov, 2004), but strongly depends on dynamic forcing (Kubryakov, Belokopytov, et al., 2019).

That is why the density rather than the depth of the mixed layer is a more robust indicator of the vertical entrainment of nutrients in winter. We use this indicator to show that in cold 2017 more nutrients are entrained in the euphotic layer than in warm 2016.

We extended the explanation in the revised version of the manuscript.

- Also, Chl-a is one of the widely-used indicators of the phytoplankton biomass, which directly depends on nutrient concentration. In 2017 Chl-a in winter and spring was higher than in 2016, which is consistent with the fact, that the winter convection and related vertical entrainment of nutrients was more intense in 2017.

We also want to underline that we are not basing on the quantitative values of nitrates, but use the above indicators to argue that in the cold winter of 2017 the vertical entrainment of nitrates was higher than in the warm winter of 2016. The increase of nutrient concentration in the Black Sea in the cold years was documented in the chemical study of (Tugrul et al., 2015).

Comment #2

"The difference between winters seems more related to mesoscale circulation than to a severe winter. In Figure 1, red squares clearly show that in winter 2017 the floats were entrapped within eddies, which would help explain also density features in Figure 2d. To support conclusion, the impact of mesoscale should be addressed and, if any, excluded".

Answer #2. The winter of 2017 was one of the most severe in the Black Sea and this fact was already documented in several recent studies by Stanev et al. (2019), Capet et al. (2020). It was significantly colder than in warm 2016 and cause significantly stronger vertical mixing than in 2016 (Stanev et al., 2019; Capet et al., 2020). It is worth noting, that in colder winters, convection will be stronger and more nutrients will be entrained in the upper layer, than in warm winter.

The investigation of the impact of small scale structures such as fronts, eddies on Chl-a is a very interesting and important problem. However, in this study, we are focused

on the annual time scales. Particularly, Fig. 5 shows that in cold 2017 Chl-a in upper layers was higher in all seasons, while in warm 2016 it was higher in deeper layers in all seasons. This fact was observed during all investigated periods of both buoys measurements. Please see Fig. 5a, which is the main figure for this manuscript. That is, yearly average profiles of Chl-a are of main interest and they depend on the intensity of winter convection (see Fig. R2 and R3 in the attached file).

The short-period variability of the Chl-a is out of the scope of this paper, but we briefly discuss it in the discussion part. Short period variability of the Chl-a in the summer period is related to the occasional entrainment of nutrients from nitrocline in the euphotic zone caused by storms or dynamical forcing (studied in the Black Sea by Kubryakov, Zatsepin et al. (2019)), such as eddies horizontal and vertical advection (see e.g. Oguz et al., 2002; Shapiro et al., 2010; Kubryakov et al., 2016). After warm winters with higher water transparency, the euphotic layer is deeper and closer to the nitrocline. Therefore, we might expect that the impact of dynamics features in summer will be more effective in years with weak winter convection.

In Fig. R2 (in the attached file) we show a diagram of 5-days averaged profiles of Chl-a for both buoys in 2016 and 2017 to demonstrate that in both years the short-period variability takes place. It is also well-seen that these two buoys were situated in different dynamic features and the Chl-a variability differs among the buoys in both years. At the same time, it is visually seen that both buoys show that in warm 2016 Chl-a subsurface maximum was deeper than in cold 2017, which is the main conclusion of the study. It is also well seen that Chl-a was higher in 2017 in the winter-spring period in upper layers and higher in summer of 2016 in deep layers (35-55 m depth) (see Fig. R2 – bottom). We will add this information in the revised version of the manuscript.

Comment #3

"Self-shading is interpreted only as a change in chlorophyll concentration (derived from fluorescence measurements) which is used as a proxy of algal biomass or at least to

indicate productive layers. Using only chlorophyll makes hard to establish if changes in the magnitude of DCM are due to actual changes in productive biomass because the same modifications could be due to photoacclimation. Optical measurements such as the optical backscattering coefficient could help decipher what changes in chlorophyll are due to actual modifications in algal biomass. Backscattering measurements are available for all the floats in the Black Sea".

Answer #3. Chl-a is the main absorbing pigments, which significantly stronger impact on the light attenuation that the backscattering by particles. Absorption is usually more than 10 times higher backscattering (see e.g. Jerlov N. G. Optical oceanography. – Elsevier, 2014). Backscattering in the Black Sea significantly depends on the intense coccolithophore blooms (nanoplankton), which despite their very strong scattering plays a minor role in the light attenuation in the sea, as well as on phytoplankton biomass (Balch, W. M., Holligan, P. M., Ackleson, S. G., & Voss, K. J. (1991). Biological and optical properties of mesoscale coccolithophore blooms in the Gulf of Maine. Limnology and Oceanography, 36(4), 629-643). The variability of backscattering in the Black Sea and its relation to coccolithophore blooms was studied in detail in our previous paper on the base of Bio-Argo data in (Kubryakov, Mikaelyan et al., 2019). In particular, in the winter-early spring season during the strongest diatom bloom, bbp does not exceed 0.002, which is only 2% of the total attenuation of PAR. During extremely strong coccolithophore blooms bbp does not exceed 0.02, which is less than 20% of attenuation coefficient. The above suggestions demonstrate that bbp is not a reliable parameter for accessing the biomass of the Black Sea phytoplankton. In our paper, we focus on the chlorophyll, which significantly control the primary productivity in the ocean.

Comment #4

"Argo data analysis lacks of details, applied protocols and procedures and this makes difficult to evaluate its appropriateness. For example, for chlorophyll, it is unclear if NPQ at the surface and CDOM influence at depth have been corrected. How chl has

been calibrated? Radiometry has been quality controlled before Kd computation?"

Answer #4. We use the standard product downloaded from ftp://ftp.ifremer.fr/ifremer/argo. The description of the used protocol is given at http://www.argodatamgt.org/Documentation. The non-photochemical quenching was not corrected. We can expect that this effect will not alter the obtained result, as we focus on the differences of Chl-a between two years. NPQ is important in the most upper layers (0-15 m), while the differences in this study were observed in all 10-70 m layer (see Fig. 5 and Fig. R2, R3). Also, NPQ depends primarily on the irradiance conditions on the surface, in which seasonal variability (change from summer to winter) is significantly larger than intraseasonal variations. Therefore, we believe the correction of NPQ will not generally change the main results of the paper.

Additional comments (AC)

AC1: "Line 15: 0.2-0.6 mg m3 is a large range of variability".

Answer AC1. We exclude these numbers from the abstract.

AC2: "Lines 17-18: to rephrase as it looks contracditory".

Answer AC2. Thank You. We rephrased it as: "These results demonstrate that the increase of light availability in warm years causes the deepening of Chl subsurface maximum and partly compensate the impact of weaker convective entrainment of nutrients on the decrease of biological productivity".

AC3: "Line 42: remove "in" before references".

Answer AC3. Removed.

AC4: "Line 59: why only these floats? More floats are available in the Black Sea".

Answer AC4. Only these floats give synchronous information on Chl-a and PAR in the study period.

AC5: "Line 60: the floats sampled a longer period. It would be interesting to see what's happened during winter in other years".

Answer AC5. To answer the question, we are sending the Chl-a variability in 3 consecutive years – warm 2016, cold 2017, and moderately warm 2018 (see Fig. R4 in the attached file). As it is seen the main differences between 2017 and 2018 are similar. In 2018 high values of Chl-a were situated deeper (up to 55 m) than in 2017 (see Fig. R4, R5, R6 in the attached file). They are smaller in the upper 0-25 m at least in the first half of the year (before the start of convection in autumn). In this paper, we are focused on the difference between consecutive cold and warm years. To avoid confusion of the reader we demonstrate these differences on the base of two contrasting years 2016 and 2017. We note, that here we use all the buoys for the analysis.

The investigation of longer Chl-a variability in 2014-2020 was made in our submitted manuscript to Progress in Oceanography. Unfortunately, it is now in the revision phase. We are sending a draft to You.

AC6: "Line 69: which equation in Xing 2011 has been used? Have you calibrated Chl with Xing procedure? In this case, I suppose you have calibrated Chl using Ed(490). Ed(490) and derived Kd are quite correlated with PAR (see Morel et al., 2007 Remote Sensing of Environment) thus making analysis of the relationships between Chl and PAR as presented in this study not fully independent".

Answer AC6. Thank You for this question. We use the standard product downloaded from ftp://ftp.ifremer.fr/ifremer/argo (Schmeichtig et al., 2015). The description of the used protocol is given at http://www.argodatamgt.org/Documentation We corrected the data description.

AC7: "Line 69: how have you taken into account spikes and other source of errors in Chl profiles?"

Answer AC7. There were no spikes in Chl-a profiles during the study period for these

two buoys. We show in Fig. R2 (in the attached file) 5-days average variability of Chl-a for both buoys in 2016-2017. The instruments work well for all the study period.

AC8: "Line 70: please give the reference for 5 umol photons".

Answer AC8. We corrected it to 2.5 $\mu$mol and provided a reference on OCR-500 documentation https://www.seabird.com/ocr-504-irradiance-cosine-response-in-water/product-downloads?id=54627923874.

AC9: "Line 73: have PAR profiles been quality controlled? Have you taken into account also clouds when computing Kd? Could it be possible to see Kd profiles? I imagine that they should be noisy as a 1 m window is very narrow".

Answer AC9. All the profiles were visually quality controlled for consistency and absence of outliers. This information was added to the text. Clouds were not taking into account. Clouds should not significantly impact on the Kd as it depends on the gradient of PAR. The Kd profiles are noisy in fact. In this study, we are focusing on monthly or even annual time scales. The statistical averaging allows us to obtain smooth results on Kd distribution. Additionally, for Fig. 2 we use smoothing by moving-average with 5 m vertical bin size. We added this information to the text. We attached below unsmoothed monthly-averaged diagrams of Kd in 2016-2017 to show the unsmoothed monthly variability and their yearly-averaged distribution (see Fig. R7 in the attached file).

AC10: "Line 77: Claustre et al 2010 is not the right reference for the public database. Argo data are available at http://doi.org/10.17882/42182 or can be downloaded from the DAC (such as Coriolis)".

Answer AC10. Thank you, it is corrected.

AC11: "Line 77: Temperature and salinity have been quality-controlled?"

Answer AC11. All the data was visually quality-controlled for consistency with known T, S distribution in the Black Sea. We added the information in the text.

AC12: "Figures 2, 4, 6: profiles have been monthly averaged? Details are missing to understand how single profiles have been managed before drawing the figures".

Answer AC12. Yes, they were monthly-averaged. We added this information to the text.

AC13: "Line 81: replace "large" with "high".

Answer AC13. Thank You. Corrected.

AC14: "Figures 2 and 3: add MLD".

Answer AC14. We added MLD to Fig. 3 (see Fig. R8 in the attached file). Despite MLD in 2017 was higher than in 2016, as it is described in the answer on comment #1 the MLD is not a proper indicator for the estimates of the intensity of entrainment of nutrients. The proper indicator is density, as nutrients are tightly related to the certain isopycnals layers.

AC15: "Line 105: to prove with data".

Answer AC15. Please, see the answer on comment #1. We extended the explanation in this part of the text.

AC16: "Line 109: why average instead of integration over depth?"

Answer AC16. This is generally similar. The use of average just seems to be more convenient for us.

AC17: "Figures 4c and d: add MLD and euphotic depth

Answer AC17. Added.

AC18: "Line 124: remove "yin yang sign".

Answer AC18. Removed.

AC19: "Line 133: add reference for isolume 3 umol photons, why not using the 1% of

irradiance just below the surface?"

Answer AC19. There is growing that the usage of the classic definition of the euphotic layer can lead to significant errors in defining productive zone (see Cullen, 2015). Euphotic layer depth is determined as the depth at which 1 % of surface PAR penetrates. Therefore, it does not take into account the impact of the strong seasonal variability of surface PAR, which can lead to significant errors in defining the depth of the productive zone (Banse, 2004). PAR at z=1 m in the Black Sea changed from 1200 to 300 $\mu$mole photons m-2 s-1. Therefore 1% of this irradiance will correspond to Ed of 12 and 3 $\mu$mole photons m-2 s-1 in summer and winter, respectively. Photosynthesis efficiency depends on the absolute values of Ed (Jassby, Platt, 1976). Phytoplankton is often observed in the layers which are significantly deeper than the depth corresponding to 1% of surface PAR (Letilier et al., 2004; Marra et al., 2014). In our submitted study (Kubryakov et al., 2020, PROOCE, submitted) we show that in the Black Sea isoline Ed of 3 $\mu$mole photons m-2 s-1 (or Qs=0.08 mole photons m-2 d-1) in all seasons correspond well to the bottom boundary of relatively high Chl-a values. Other authors often use other definitions, e.g. (Mayot et al., 2017) use 0.415 mol photons m-2 d-1. Therefore, the usage of the value of Ed=3 $\mu$mole photons m-2 s-1 for the definition of the lower border of the productive zone in the Black Sea intuitively is less robust. We are sending Your submitted manuscript (Kubryakov et al., 2020, PROOCE, submitted) for more details.

AC20: "Line 134: shallower instead of higher?"

Answer AC20. Corrected.

AC21: "Figures 5 and 6: I suggest to show PAR and Kd time series on the same figure".

Answer AC21. We agree and corrected the text.

AC22: "Caption figure 6: explain the choice of 25 umol photons".

Answer AC22. We deleted this isolume to avoid confusion.

AC23: "Figure 6: show MLD, DCM, euphotic depth".

Answer AC23. We added euphotic depth to this figure. MLD is shown in Fig. 2a,b. It is not clear how to show DCM in this figure. Instead, we show isolumes in Fig. 2a, b.

AC24: "Figure 7 does not sum up your case study as in the figure MLDs are different. In this study the MLDs in winter look very similar".

Answer AC24. We agree with this comment. As it is stated in the answer on comment #1: "At the same time, the mixed layer depth in the cold period of a year may vary significantly due to the dynamical forcing, such as eddies, large-scale circulation e.t.c. (see in details in Kubrykov et al., 2019). This is related to the deepening of the density barrier – the main halocline. For example, in anticyclones, it can reach 100 m. However, if the density of the mixed layer remains low, then no new nitrates will be entrained from deep isopycnals layers.

The density of the mixed layer depends partly on the vertical uplift of isopycnals during the intensification of cyclonic circulation. Rise of cyclonic circulation on opposite decrease mixed layer depth. Therefore, in the Black Sea the MLD is not correlated with sea surface temperature (Titov, 2004), but strongly depends on dynamic forcing (Kubryakov, Belokopytov, et al., 2019). That is why the density rather than the depth of the mixed layer is a more robust indicator of the vertical entrainment of nutrients in winter". To account for this fact, we change the label of the figure to "winter mixing".

AC25: "Paragraph at Line 208: the influence of CDOM is less detectable from PAR as PAR integrates irradiance between 400 and 700 nm while the highest CDOM influence is in the UV range. As irradiance profiles at 380 nm are measured by BGC floats, Kd(380) should be computed to corroborate or not your statements".

Answer AC25. The detailed investigation of the variability of Kd(380), as well as DOM on the base of its measurements, were made in our previous study (Kubryakov, Mikaelyan, et al., 2019). We have extended the explanation in this part of the text:

"Another strong source of light attenuation, which is nutrient-dependent, is autochthonous DOM formed due to the release of lipids during lysis of phytoplankton cells. In (Kubryakov, Mikaeyan et al., 2019) based on the Bio-Argo data on diffuse attenuation at 390 nm and backscattering measurement authors shows that one the most significant source of DOM is related to the lysis of coccolithophores cells during the termination of their early-summer bloom. As it is shown in (Burenkov et al., 2011; Mikaelyan et al., 2011, 2015) the intensity of coccolithophore blooms in the Black Sea are significantly related to winter temperature and amount of entrained nutrients. Particularly, the extremely strong coccolithophore bloom was observed in the Black Sea after cold 2017, which results in the observed maximum light attenuation in July-August in 2017 at 10-30 m depth (Fig. 5b). Such DOM release in cold years additionally increases the light attenuation, which plays a role in equalizing the total productivity in warm and cold years".

AC26: "Line 213: please give some explanations".

Answer AC26. We corrected this phrase.

AC27: "Line 214: Are lipids colored?"

Answer AC27. This depends on the lipids content. The one discussed here causes strong light attenuation at 390 nm, as it is in detail investigated by Kubryakov, Mikaelyan, et al. (2019), i.e. are partly colored. We extended the explanation in this part of the text (see the answer to the AC25).

AC28: "Line 237: with no nutrient data this statement cannot be proved by this study".

Answer AC28. We agree and excluded this phrase from the text. As it is explained in the answer on comment #1, in this paper we can use only indirect estimates of the intensity of the vertical entrainment of nutrients. Previously, the increase of nutrient concentration in the Black Sea in the cold period was documented in the chemical study by Tugrul et al. (2015).

Please also note the supplement to this comment:
https://www.biogeosciences-discuss.net/bg-2020-210/bg-2020-210-AC2-supplement.pdf

——————————————

[Figure]

**Fig. 1.** Fig. R1: left– Interannual diagram of NO3 ($\mu$M) measured by buoy Bio-Argo; right– mul-
tiannual averaged vertical profiles of NO3 ($\mu$M, black line) and PO4 ($\mu$M, red line) for October
shown in $\sigma$-coordinate

[Figure]

**Fig. 2.** R2.Time variability of Chl by Bio-Argo buoy measurements #7900591(top) and buoy #6901866(central) in 2016(left) and 2017(right).Bottom–differences between 2017,2016 by buoy #6901866(left), #7900591(ri

[Figure]

**Fig. 3.** Fig. R3. Average profile of Chl-a in 2016 and 2017 by the measurements of the buoy #6901866 (left) and buoy #7900591(center) and both buoy measurements (right).

[Figure]

**Fig. 4.** Fig. R4. Seasonal diagrams of Chl-a in 2016, 2017, and 2018 years from Bio-Argo data.

[Figure]

**Fig. 5.** Fig. R5. Difference of monthly averaged diagram of Chl-a between 2017 and 2016 (left), 2017 and 2018 (right).

[Figure]

**Fig. 6.** Fig. R6. Average profile of Chl in 2016 and 2017 (left), 2017 and 2018 (right) by Bio-Argo buoy measurements (right).

[Figure]

**Fig. 7.** Fig. R7. Monthly-averaged vertical distribution of Kd (PAR) in 2016 (left) and 2017(center), Kd yearly-averaged profile in 2016 and 2017 (right) in the Black Sea from Bio-Argo data.

[Figure]

**Fig. 8.** Fig. R8. Average seasonal variability of Chl, temperature and mixed layer depth in 2016
(solid line) and 2017 (dashed) in the 0-70 m layer.